# Induced Pluripotent Stem Cells-Based Regenerative Therapies in Treating Human Aging-Related Functional Decline and Diseases

**DOI:** 10.3390/cells14080619

**Published:** 2025-04-21

**Authors:** Peijie Yu, Bin Liu, Cheng Dong, Yun Chang

**Affiliations:** 1Department of Biomedical Engineering, The Hong Kong Polytechnic University, Hunghom, Hong Kong 999077, China; peijie.yu@connect.polyu.hk (P.Y.); bin-bme.liu@polyu.edu.hk (B.L.); 2The Hong Kong Polytechnic University Shenzhen Research Institute, Shenzhen 518057, China

**Keywords:** induced pluripotent stem cells, aging-related diseases, regenerative medicine, cellular reprogramming, personalized therapies

## Abstract

A significant increase in life expectancy worldwide has resulted in a growing aging population, accompanied by a rise in aging-related diseases that pose substantial societal, economic, and medical challenges. This trend has prompted extensive efforts within many scientific and medical communities to develop and enhance therapies aimed at delaying aging processes, mitigating aging-related functional decline, and addressing aging-associated diseases to extend health span. Research in aging biology has focused on unraveling various biochemical and genetic pathways contributing to aging-related changes, including genomic instability, telomere shortening, and cellular senescence. The advent of induced pluripotent stem cells (iPSCs), derived through reprogramming human somatic cells, has revolutionized disease modeling and understanding in humans by addressing the limitations of conventional animal models and primary human cells. iPSCs offer significant advantages over other pluripotent stem cells, such as embryonic stem cells, as they can be obtained without the need for embryo destruction and are not restricted by the availability of healthy donors or patients. These attributes position iPSC technology as a promising avenue for modeling and deciphering mechanisms that underlie aging and associated diseases, as well as for studying drug effects. Moreover, iPSCs exhibit remarkable versatility in differentiating into diverse cell types, making them a promising tool for personalized regenerative therapies aimed at replacing aged or damaged cells with healthy, functional equivalents. This review explores the breadth of research in iPSC-based regenerative therapies and their potential applications in addressing a spectrum of aging-related conditions.

## 1. Introduction

Aging is currently one of the most significant demographic trends globally, with profound implications for social, economic, and medical systems [1,2,3]. According to the World Health Organization, the proportion of people aged 60 and older is expected to nearly double from 12% in 2015 to 22% in 2050, with an absolute increase from 900 million to 2 billion [3]. Aging is a significant risk factor for cardiovascular diseases, musculoskeletal issues, neurological disorders, immunosenescence, and increased mortality or multimorbidity induced by various chronic diseases [4,5,6,7,8]. In response to these challenges, scientists from different backgrounds have explored a wide array of strategies to extend healthy lifespans, including lifestyle modifications, pharmacological and genetic manipulations, and advanced medical interventions [9]. Central to these efforts is the study of the molecular and cellular processes that drive aging, such as genomic instability, telomere attrition, and cellular senescence.

However, the heterogeneity of aging manifestations makes it difficult to pinpoint specific mechanisms and develop targeted medical interventions, and thus a diverse range of therapeutic interventions is still being explored and developed, including stem cell therapy. Stem cell theory is a broad concept in biology and medicine that revolves around many unique properties and functions of stem cells, which are undifferentiated cells capable of dividing and differentiating into specialized cell types [10,11]. This theory has significant implications across various fields, including regenerative medicine, cancer biology, and developmental biology. However, traditional stem cell therapies still face some obstacles, including ethical concerns, immunological rejection, restricted cell source availability, and limited differentiation potential [12]. In recent years, active research has focused on iPSCs, which are a type of stem cell that can be generated directly from adult cells. This groundbreaking technology, first demonstrated by Shinya Yamanaka and colleagues in 2006, involves reprogramming adult somatic cells into a pluripotent state by introducing specific transcription factors [13,14]. iPSCs possess a unique ability to differentiate into a wide array of cell types, making them invaluable for creating models of human aging diseases and for testing potential therapeutic interventions [12]. Furthermore, their capacity for patient-specific derivation allows for the exploration of genetic and environmental factors that contribute to individual aging processes and disease susceptibilities [12].

Taking into account the extensive application prospects of iPSCs, this review delves into the current state of research on iPSCs-based regenerative therapies, focusing on their application in understanding and treating aging-related diseases. We will explore the molecular mechanisms underlying human aging-related functional decline and diseases, the processes of cellular reprogramming for iPSCs generation, and the use of iPSCs-based models to develop the therapeutic potentials of iPSC-derived differentiated cells in combating these conditions. Through this exploration, we aim to highlight the transformative potentials of iPSC technology in addressing the multifaceted challenges posed by aging and aging-related societal, economic, and medical issues.

## 2. Molecular Mechanisms of Aging Diseases

Aging is an inherent biological process characterized by a gradual and irreversible deterioration in many physical functions across all organ systems, primarily caused by the accumulation of damage in response to various stressors. As a result, aging leads to a range of conditions that cause functional decline and increase susceptibility to health complications. The hallmarks of aging-related diseases include a series of interconnected processes that contribute to the gradual breakdown of tissue function and overall health as people age [15]. These hallmarks are broadly classified into tissue, cellular, and molecular mechanisms, which are interrelated and jointly interconnected, and contribute to the overall aging phenotype (Figure 1).

### 2.1. Tissue Level Hallmarks of Aging

Both stem cell exhaustion and the loss of proteostasis are critical hallmarks of aging that contribute to the decline in tissue function and regenerative capacity. Understanding these mechanisms provides potential targets for therapeutic interventions aimed at enhancing stem cell behavior and restoring proteostasis to improve tissue health and extend lifespan (Figure 2).

#### 2.1.1. Stem Cell Exhaustion

Stem cell exhaustion refers to the decline in the number and functionality of stem cells with aging, which impairs tissue regeneration and maintenance [16]. In humans, the ages of both the donor and recipient affect the outcomes of hematopoietic stem cell transplants, highlighting that stem cell function diminishes with age [16]. This decline plays a significant role in the degeneration of tissues such as muscle, skin, and bone commonly seen with aging [17]. For example, with aging, the regenerative capacity of various stem cells—such as muscle stem cells, hair follicle stem cells, and bone marrow mesenchymal stem cells—decreases. This reduction leads to conditions like sarcopenia, osteoporosis, and a decreased ability to heal wounds, along with an increased risk of chronic wounds [18,19,20]. To be more specific, aged muscle stem cells produce less keratocan, disrupting the ECM structure necessary for muscle integrity [21]. Furthermore, single-cell RNA sequencing combined with functional studies in mouse models has shown that while aged hair follicle stem cells remain undifferentiated, they lose some of their lineage-specific functions [18]. Aging also reduces the pool of BM-MSCs and biases their differentiation towards adipocytes rather than osteoblasts, contributing to osteoporosis [19]. Thus, developing therapies using younger or rejuvenated stem cells may mitigate aging-related diseases and improve tissue regeneration.

#### 2.1.2. Loss of Proteostasis

Proteostasis, or protein homeostasis, refers to the maintenance of the integrity and functionality of the proteome through regulatory networks that control protein synthesis, folding, and degradation [22]. With aging, there is an increase in oxidative damage that accumulates randomly in proteins, leading to misfolding and aggregation. This disrupts healthy synthesis and degradation processes, overwhelming the capacity of chaperones and causing cellular dysfunction [22]. Wrongly folded proteins can accumulate and form toxic aggregates that can disrupt the normal functioning of cellular organelles, especially the endoplasmic reticulum and mitochondria, leading to cellular dysfunction and contributing to neurodegenerative disorders such as Alzheimer’s disease [23,24]. On the other hand, the ubiquitin–proteasome system and autophagy–lysosome pathways are crucial for degrading damaged proteins. These systems become less efficient, leading to the accumulation of misfolded proteins as one gets older [25]. This decline in proteostasis is also closely linked to age-related muscle wasting. Interestingly, while the overall activity of the mammalian target of mTOR complex1 may decrease, its activation in specific muscle fibers can result in morphologically abnormal mitochondria and increased oxidative stress [26]. In addition, the neuromuscular junction is the connection between nerves and muscles, and its deterioration as one becomes older can disrupt signaling, leading to impaired muscle function and strength [27].

### 2.2. Hallmarks at the Cellular Level

The aging process is driven by several interconnected mechanisms, among which genomic instability, telomere attrition, epigenetic alterations, and cellular senescence play pivotal roles (Figure 3).

#### 2.2.1. Genomic Instability

Genomic instability is one of the primary hallmarks of aging. It refers to an increased rate of DNA damage and mutations that accumulate over time, leading to errors in DNA replication and repair mechanisms [28]. The “DNA damage accumulation theory of aging” posits that unrepaired DNA damage plays a central role in promoting aging [28]. DNA damage can originate from both endogenous sources (e.g., oxidative stress, replication errors) and exogenous sources (e.g., UV radiation, chemicals) [29]. Cells have evolved intricate DNA damage response pathways to detect and repair DNA damage. However, mutations in genes involved in DNA damage response can impair these pathways, leading to an accumulation of unrepaired DNA lesions [30]. If the damage is too severe, cells may undergo mitotic catastrophe, resulting in tetraploid cells with abnormal nuclear morphologies [31]. This instability increases the mutation load and genetic heterogeneity within cellular populations, contributing to various chronic conditions, especially to the aging-related diseases like cancer and neurodegeneration. The increased expression of mismatch repair proteins, which usually correct errors during DNA replication, has been linked with tumor aggressiveness when overexpressed, indicating that even mechanisms meant to maintain stability can paradoxically lead to instability under certain conditions [32]. Regarding neurodegenerative diseases, chromosome instability seems to play a significant part in the pathogenic cascades leading to neurodegeneration in late adulthood [33]. The brain-specific manifestation of chromosome instability appears to be particularly important for these diseases. Additionally, somatic mutations accumulate in neurons over time, potentially driving brain aging and neurodegeneration [34]. Single-cell whole-genome sequencing studies have identified somatic single-nucleotide variants in neurons from individuals affected by early-onset neurodegenerative diseases, supporting this notion [34].

#### 2.2.2. Telomere Attrition

Telomeres, the protective structures at the ends of chromosomes, shorten progressively as cells divide. When they become critically short, cells enter a state known as senescence [35]. In addition, the telomeres in nonimmune cells are highly susceptible to oxidative damage induced by neighboring neutrophils. Studies have demonstrated that neutrophils may cause telomere dysfunction in vitro and ex vivo in a ROS-dependent manner [36]. Critically short or dysfunctional telomeres can lead to chromosomal instability, such as end-to-end fusions, which may result in genetic abnormalities and increase the risk of cancer [37]. The gradual shortening of telomeres with each cell division leads to cellular dysfunction, which is commonly observed in aging-related conditions such as cardiovascular disease and pulmonary fibrosis [38]. For example, one study had shown that mice lacking telomerase in endothelial cells exhibit senescence and hypoxia-induced dysfunction, leading to accelerated aging and pathogenesis in various organs, including the heart [39]. The dysfunction of telomeres in endothelial cells exacerbates several signs of vascular and metabolic aging, including impairments in blood vessel function and glucose tolerance [40]. For pulmonary fibrosis, particularly IPF, telomere dysfunction in alveolar type II cells has been shown to lead to lung fibrosis. The deletion of the telomere shelterin protein TRF1 in these cells causes replicative senescence and inflammatory phenotypes [41]. Genetic alterations in telomere maintenance genes are also linked to familial and sporadic IPF [42]. The reason for this may be that telomere shortening in alveolar type II cells and endothelial cells leads to impaired regenerative capacity and increased susceptibility to fibrotic changes in the lungs [43].

#### 2.2.3. Epigenetic Alterations

Epigenetic changes, such as DNA methylation and histone modification, regulate gene expression without altering the DNA sequence. Normally, DNA methylation is highly specific and stable. Adequate levels of DNA methylation are necessary for healthy aging and ensure that certain genes remain silenced while others are expressed appropriately [44]. However, with aging, these patterns degrade, leading to the loss of the specificity and random variations in gene expression that can impair cellular function [45]. This increase in variability in DNA methylation levels across individuals, known as ‘epigenetic drift’, can result in more randomness in gene expression profiles [46]. Conversely, hypermethylation (excessive methylation) in regions that are usually unmethylated, such as CpG islands within promoters, can silence important regulatory genes, disrupting normal cellular functions like critical epithelial–mesenchymal transition [47]. These modifications play a pivotal role in aging-related diseases like osteoporosis metabolic disorders and fibrosis [48,49,50,51]. For instance, the chromatin remodeling factor Arid1a plays a regulatory role in the differentiation of osteoclast precursors [52]. The deletion of the chromatin remodeling factor Arid1a leads to chromatin reprogramming, which restricts access to promoters by transcription factors such as Jun/Fos, thereby inhibiting osteoclast activation and bone resorption [52]. In addition, the hypermethylation of genes involved in β-cell differentiation and function can lead to impaired insulin secretion, contributing to the development of T2DM [53,54]. A recent study also found that the hypermethylation of Hoxa5 at its gene promoter in mouse unilateral ureteral obstruction kidneys also results in the deletion of Hoxa5, which promotes fibrosis in renal disease through the induction of JAG1 and the subsequent activation of the Notch signaling pathway [55].

#### 2.2.4. Cellular Senescence

Senescent cells, which are characterized by a permanent cell cycle arrest, accumulate in tissues over time and contribute significantly to aging and aging-related diseases [56,57]. Senescent cells accumulate over time and secrete pro-inflammatory factors, contributing to chronic inflammation and tissue dysfunction. One key mechanism through which these cells exert their effects is via the secretion of pro-inflammatory molecules, including the SASP, a collection of secreted factors including inflammatory cytokines, chemokines, growth factors, and proteases [58,59,60]. These factors not only induce inflammation but also promote the senescence of neighboring cells in a paracrine manner, leading to a vicious cycle that amplifies the inflammatory response and contributes to tissue dysfunction [61,62]. In the circulatory system, research on endothelial cells reveals that senescent vascular endothelial cells often lead to chronic inflammation and oxidative stress, which in turn induces endothelial dysfunction and contributes to the development of atherosclerosis [63]. Senescent vascular smooth muscle cells also accumulate in aged arteries and contribute to vascular stiffness and dysfunction through the interleukin-1α-driven SASP and the priming of adjacent cells to a proatherosclerotic state [64]. Senescent RPE cells exhibit impaired lysosomal function, which affects their ability to degrade photoreceptor outer segments and other cellular waste products like lipofuscin [65]. As a result, in the process of AMD, senescent RPE cells and choroidal cells have been found to contribute to retinal dysfunction by inducing local retinal swelling and the disorganization of retinal layers [66]. For arthritis, SASP exacerbates joint destruction by promoting the release of MMPs and other destructive enzymes. In osteoarthritis, chondrocytes undergo hypertrophic and senescent transitions, leading to the suppressed expression of key cartilage matrix genes and the increased production of inflammatory mediators [67]. Similarly, in rheumatoid arthritis, senescent cells also play a role by potentially modulating immune responses and contributing to chronic inflammation [68]. In a word, senescence is implicated in various aging-related diseases, and targeting senescent cells and their SASP offers a promising therapeutic approach to mitigate these conditions.

These processes are not isolated, but rather form a complex network that is mutually reinforcing. For instance, genomic instability can accelerate telomere shortening, while telomere dysfunction can induce DNA damage and senescence. Meanwhile, epigenetic changes can influence both the stability of the genome and the regulation of senescence-related genes [69]. The accumulation of senescent cells further promotes genomic instability and epigenetic alterations through their SASP. For instance, IL-6 has been shown to activate the cGAS–STING–NFκB pathway, leading to increased cytosolic DNA and subsequent genomic instability [62]. HMGB1 release from senescent cells can also promote chromatin remodeling [70].

### 2.3. Molecular Level Hallmarks

Mitochondrial dysfunction, dysregulated nutrient sensing, and inflammatory signaling are interconnected and collectively drive the aging process. They affect multiple tissues and organs, contributing to the general decline in physiological function observed in aging individuals (Figure 4).

#### 2.3.1. Mitochondrial Dysfunction

Mitochondria are the primary organelles responsible for producing ATP, the energy currency of the cell [71]. When mitochondrial function is compromised, ATP synthesis becomes less efficient, resulting in not only reduced energy output, but also increased ROS production, disrupted protein homeostasis, and impaired cellular quality control [71]. Under normal conditions, cells maintain a balance between ROS production and scavenging through an intricate antioxidant system. However, this equilibrium can be disrupted when ROS levels rise excessively due to environmental stressors, metabolic overload, or genetic factors [72]. mtDNA is especially susceptible to ROS-induced damage because, unlike nuclear DNA, it lacks protective histones and has inefficient repair mechanisms. Excessive ROS can cause mutations and deletions in mtDNA, affecting the expression of essential respiratory chain subunits [73]. This damage accumulates over time, leading to impaired electron transport chain function and further ROS production [74,75]. Studies have shown that mitochondrial dysfunction in aging tissues results in less ATP production while increasing the production of ROS as byproducts of aerobic metabolism [76]. The antioxidant defense system—including enzymes like superoxide dismutase, catalase, and glutathione peroxidase—may become overwhelmed, failing to neutralize all ROS and resulting in oxidative damage to vital mitochondrial components [77,78]. In summary, the cumulative damage to mtDNA, lipids, and proteins disrupts mitochondrial homeostasis. Dysfunctional ETC complexes produce even more ROS, creating a vicious cycle of oxidative damage and functional decline [75]. These factors link significantly to aging-related conditions like neurodegenerative diseases and metabolic disorders. In Alzheimer’s and Parkinson’s diseases, for example, neurons—which have high energy demands—are particularly vulnerable to reduced ATP production and increased ROS. Restoring energy supply and reducing oxidative stress may help mitigate neuronal dysfunction [79,80,81]. Mitochondrial dysfunction also plays a key role in metabolic disorders like T2DM. Insulin helps maintain mitochondrial proteome integrity, but this regulation is impaired in metabolic diseases [82]. In individuals with T2DM, there is a significant reduction in mitochondrial OXPHOS capacity in insulin-sensitive tissues like skeletal muscle, liver, and adipose tissue [83]. In addition, mitochondrial dysfunction in cardiac tissue is associated with reduced mitochondrial capacity and coupling efficiency, contributing to diabetic cardiomyopathy [84].

#### 2.3.2. Dysregulated Nutrient Sensing

Nutrient-sensing pathways like mTOR, AMPK, and sirtuins regulate cellular metabolism and growth in response to nutrient availability. The dysregulation of these pathways can lead to metabolic disorders and accelerated aging. mTOR, a key regulator of cell growth and metabolism, tends to become overactive as one becomes older, promoting anabolism at the expense of catabolic processes necessary for cellular maintenance [85]. Conversely, AMPK, which detects low energy states, becomes less efficient over time, weakening the cell’s ability to cope with metabolic stress [86]. Sirtuins, a family of NAD+-dependent deacylases, act as metabolic sensors that respond to the intracellular NAD+/NADH ratio, which fluctuates with nutrient levels [87]. They facilitate DNA damage repair, postpone telomere shortening, and induce longevity benefits of heat restriction [88]. Among them, SIRT1 is particularly important, as it helps delay cellular senescence and boosts stress resistance [89,90]. However, age-related declines in SIRT1 levels contribute to the buildup of senescent cells [90]. The insulin/IGF-1 signaling pathway is another critical regulator of glucose metabolism. Reduced signaling through this pathway has been associated with improved insulin sensitivity, metabolic health and even extended lifespan in various models [91,92]. However, the effects of IGF-1 modulation on aging are complex and context-dependent. Controversial lifespan data have emerged from studies using different methods to modify IGF-1 or its receptor expression in mice [93]. Studies focusing on cardiac aging also found that the overexpression of IGF-1 improved cardiomyocyte contractile function in old mice [94], whereas liver-specific IGF-1 deficiency led to improved hepatic function and autophagy in aged mice [95]. These discrepancies underscore the tissue-specific and experimental nuances of IGF-1’s role in aging. Importantly, IGF-1 remains vital for normal development and tissue maintenance throughout life, suggesting that its effects on aging are multifaceted—balancing benefits in one context against potential drawbacks in another [96].

mTOR and IIS pathways are closely intertwined, often working together to coordinate cellular responses to nutrients. For instance, Crosby et al. demonstrated that IGF-1 helps align metabolism with circadian rhythms by boosting PERIOD clock protein expression, a process dependent on mTORC1 as a critical signaling hub [97]. Conversely, reduced signaling through the insulin/IGF-1 pathway can inhibit mTORC1, leading to enhanced autophagy, improved metabolic health, and even prolonged lifespan in mice [98]. In neurodegenerative diseases like Alzheimer’s, chronic mTOR overactivation disrupts insulin signaling by inhibiting IRS1, further impairing metabolic regulation [99]. This interplay highlights how nutrient-sensing pathways collectively influence aging, metabolism, and disease.

#### 2.3.3. Inflammatory Signaling

Chronic low-grade inflammation, known as “inflammaging” or “senoinflammation”, is a key feature of aging that drives the onset and progression of age-related diseases [100,101]. This condition is characterized by an unresolved and low-grade inflammatory process that impacts multiple organs and systemic functions, leading to progressive degeneration over time. The exact mechanism behind inflammation is multifaceted and it can be fueled by a variety of pro-inflammatory stimuli, which involve several key processes, such as senescence and immune system dysregulation. In summary, the immune system undergoes significant changes as age increases, a phenomenon known as immunosenescence [102]. This includes a decline in adaptive immunity and increased activation of the cGAS-STING pathway by innate immune activation, leading to a pro-inflammatory environment [103]. In fact, immunosenescence and aging-related inflammation drive each other, and vice versa, contributing to a vicious cycle. This sustained inflammation heightens susceptibility to cardiovascular diseases in older adults, accelerating conditions like atherosclerosis and heart failure [104]. In the brain, inflammaging worsens protein aggregation pathologies. For example, preformed amyloid-β fibrils can also cross-seed the tau protein, driving its aggregation into NFTs [105]. Inflammaging not only speeds up the deposition of these misfolded proteins but also amplifies their neurotoxicity, leading to synaptic damage, neuronal death, and cognitive impairment [106].

Among these closely linked markers, emerging research highlights cross-talk between nutrient-sensing pathways and inflammation. For example, even the modest overactivation of mTOR signaling can lead to the increased production of pro-inflammatory cytokines [107]. In specific contexts, such as in myeloid dendritic cells, mTOR inhibitors have been observed to increase the expression of pro-inflammatory cytokines [108]. Similarly, microglia—the brain’s resident immune cells—enter a chronic activated state during inflammaging, releasing harmful cytokines and ROS that exacerbate neuronal damage and protein aggregation [109].

Understanding these hallmarks of aging diseases at the tissue, cellular, and molecular levels provides insights into the complex interplay of factors contributing to aging-related functional decline and disease susceptibility. Targeting these mechanisms through iPSCs-based intervention holds great promise for promoting healthy aging and mitigating the impact of aging-related diseases.

## 3. Cellular Reprogramming for Induced Pluripotent Stem Cell Generation

Before the groundbreaking research that led to the induction of pluripotent stages in somatic cells through Yamanaka’s four transcription factors (Oct4, Sox2, Klf4, and c-Myc) [14], embryonic stem cells (ESCs) derived from the inner cell mass of a blastocyst were regarded as a primary source of pluripotent cells [110,111]. These ESCs demonstrated remarkable potential, capable of differentiating into all three germ layers while maintaining genomic stability during expansion, making them invaluable for regenerative medicine research [112,113,114,115]. However, ESC applications faced significant limitations, including ethical controversies surrounding embryo use, immune rejection risks, difficulties in maintaining homogeneous cell populations during differentiation, and tumor formation potential [116,117,118]. Additionally, the derivation of ESCs involves the destruction of human embryos, which has raised ethical issues [119].

Cellular reprogramming emerged as a transformative solution, with Yamanaka’s 2006 breakthrough enabling the creation of iPSCs from mature somatic cells (Figure 5). This landmark achievement demonstrated that specialized cells like skin or blood cells could be reverted to a pluripotent state resembling ESCs through the introduction of just four transcription factors [14]. This process involves resetting the cellular identity of specialized cells like skin or blood cells back to a pluripotent state akin to embryonic stem cells. The key steps and mechanisms involved in cellular reprogramming for iPSC generation can be succinctly summarized.

### 3.1. Introduction of Reprogramming Factors

In fact, the expressions of Oct4, Sox2, Klf4, and c-Myc (OSKM factors) have been found to promote tumorigenesis in previous studies. Their elevated expression is observed in various cancers, including cervical cancer and liver cancer [123,124,125]. Furthermore, the OSKM factors can contribute to the maintenance of a subpopulation of cancer cells with stem-like properties, which drives tumor progression and therapy resistance [126].

Although OSKM factors are associated with tumor formation, their application under specific conditions offers great potential and benefits for scientific research and medicine. The process of reprogramming is initiated by introducing specific reprogramming factors into the target cells. These factors can activate or repress key genes responsible for maintaining pluripotency and play a key role in the reprogramming process [13]. Researchers have explored the effects of introducing OSKM factors into specific tissues, such as the heart, to observe how cardiac cells respond to the ectopic expression of these factors [127]. The direct intramyocardial injection of non-integrating adenoviral vectors expressing OSKM was used to investigate reprogramming in vivo, but it showed that there was no significant regenerative effect after myocardial infarction [127]. The mouse cell line established by Mao et al. consistently induces OSKM expression to derive and maintain pluripotent cells without the use of specific growth factors or signaling inhibitors that are typically required in conventional culture systems. Importantly, further experiments injecting iPSC generated by sustained OSKM expression into mouse blastocysts revealed that OSKM-iPSC contributed to various organs and tissues after the formation of chimeric embryos. This study showed that OSKM-induced iPSCs have the potential for application in other species to generate genetically modified animals through lineage transmission [128].

To enhance reprogramming efficiency, researchers have identified factors that can significantly boost the efficiency of iPSCs generation. For instance, p53 siRNA and UTF1 (Undifferentiated Embryonic Cell Transcription Factor 1) were found to enhance reprogramming efficiency up to 100-fold when used alongside OCT4, SOX2, KLF4, and c-MYC, even without the oncogene c-MYC [129].

### 3.2. Initiation of Dedifferentiation and Transition to a Pluripotent State

The reprogramming factors synergistically induce a dedifferentiation process in the target cells, erasing the specialized identity of cells, and reversing the epigenetic modifications that define their differentiated state. The directed differentiation of iPSCs into specific cell types is critical for their therapeutic utility. Nanotopography platforms have been utilized to study iPSCs differentiation, revealing gene expression profiles associated with poor differentiation outcomes [130,131]. Conclusions drawn from these studies emphasize the importance of carefully selecting and designing nanotopographical surfaces for guiding iPSCs differentiation towards more predictable and efficient results. Furthermore, they highlighted the potential role of mechanical factors, such as substrate stiffness, in influencing cellular responses during differentiation. As cells undergo dedifferentiation, they experience profound genomic and epigenetic reorganization, including large-scale gene expression changes and chromatin remodeling. This process culminates in the acquisition of pluripotency, characterized by unlimited self-renewal capacity and the potential to differentiate into any cell type derived from the three germ layers (ectoderm, mesoderm, and endoderm).

### 3.3. Verification of Pluripotency

After reprogramming, iPSCs undergo characterization to assess their pluripotency via molecular markers, gene expression profiling, and functional assays [132]. These cells should exhibit pluripotent properties similar to those of embryonic stem cells, including the ability to form teratomas in vivo, as well as to differentiate into cell types of all three germ layers in vitro and to offer validation against these characteristics [133,134]. Modern techniques now enable the comprehensive verification of iPSC pluripotency, including the detailed analysis of DNA methylation patterns. Ping, W.; et al. compared the genome-wide DNA methylation patterns of mouse chemically produced iPSC, OSKM-integrated iPSCs, and mouse ESCs [135]. All three types of pluripotent stem cells tend to have low overall DNA methylation levels, but there were cell type-specific differences in certain regions, such as retrotransposons. Chemically produced iPSCs presented closer epigenetic features to mouse ESCs compared to OSKM-integrated iPSCs [135].

Since their discovery two decades ago, iPSC generation has experienced exponential growth in research applications. However, the extended timeline required for functional human neuronal differentiation, coupled with the constraints of conventional 2D culture systems, continues to present substantial hurdles for disease modeling and therapeutic development. To address these challenges, researchers have turned to innovative approaches like conductive graphene scaffolds, which enhance neuronal differentiation by simultaneously providing mechanical support and electrical stimulation [136].

Additionally, scientists around the world have tried various methods to avoid insertion mutagenesis, a major limitation of the retroviral approach, but no technology had been found that can completely avoid DNA integration at first. Tavernier, G.; et al. successfully activated pluripotency-related genes in mouse embryonic fibroblasts by the cationic lipid-mediated introduction of mRNA encoding these four factors, greatly reducing the number of transfections required [137]. Recently, non-integrating methods such as Sendai virus and episomal vectors have also been developed [138]. As an RNA virus, Sendai virus replicates in the cytoplasm and does not integrate into the host cell’s genome, thus avoiding insertional mutagenesis [139]. Episomal vectors are maintained as extrachromosomal elements and are gradually lost during cell division, and thus offer a safer alternative to integrating viral vectors, reducing the risk of insertional mutagenesis and oncogene activation [138]. These methods allow for the delivery of reprogramming factors without integrating into the host genome, thereby reducing the risk of insertional mutagenesis.

Importantly, due to the potential carcinogenic risk of OSKM, there are ongoing efforts to replace or supplement these factors with extracellular signals and microenvironmental cues, especially those synthetic biomaterials. The stiffness of the substrate was found to guide stem cell differentiation into specific lineages, such as neurons or muscle cells, while softer substrates have been shown to improve human cell reprogramming outcomes compared to stiffer ones [140,141,142]. For example, Chowdhury et al. found that a novel reprogramming regulator—protein phosphatase and actin regulatory factor 3—is upregulated at very early time points under gel conditions, which accounts for the enhanced reprogramming results observed [142]. In another case assessing biochemical cues, small molecules enabled the direct reprogramming of rat cardiac fibroblasts into cardiomyocyte-like cells, exhibiting typical molecular cardiac markers, morphological characteristics, and calcium flux activity [143]. The combination of mechanical and biochemical cues in synthetic biomaterials can significantly influence cell reprogramming. Recently, Li, S.; et al. demonstrated that polydimethylsiloxane, a hydrophobic and soft substrate, facilitates adherent spheroid formation, promoting cellular physical reprogramming into stem-like cells without the need for transcription factors [144].

Synthetic DNA combined with cell surface engineering can also address limitations related to iPSC genomic stability and reprogramming efficiency. For instance, introducing synthetic mRNA encoding CYCLIN D1, a protein involved in homologous recombination, has been shown to enhance DNA repair during reprogramming, leading to genetically stable iPSCs [145]. Importantly, Wang, X.; et al. found that the physical method of embedding DNA into the cell membrane by combining it with lipids is faster and more efficient. The chemical reaction that covalently binds DNA to the cell surface can make the survival time (half-life) of DNA on the cell surface 3–4 times longer than that of DNA connected by physical methods, which has guiding significance for the surface engineering of iPSCs [146].

The transition to large-scale iPSC manufacturing represents a critical step for clinical translation, yet significant technical challenges remain. Current limitations in suspension bioreactor systems, particularly regarding efficient nutrient delivery and metabolic waste removal, must be overcome to enable industrial-scale production. A promising Biological System-of-Systems framework has recently been developed to address these challenges by modeling complex cell-to-cell interactions and optimizing nutrient distribution [147]. This innovative approach features a modular design that integrates multiple data streams, allowing for the precise characterization of cellular interactions across different scales—from single cells to entire populations—while improving predictions for both layered and aggregate culture systems [147]. Complementing this advancement, Ackermann, M.; et al. established robust protocols for scalable iPSC expansion using orbital shaker and stirred-tank bioreactor platforms, specifically demonstrating their utility for the mass production of iPSC-derived macrophages [148].

## 4. Induced Pluripotent Stem Cell-Based Models for Human Aging Diseases Investigation

The unique combination of self-renewal capacity and pluripotency has positioned iPSCs as transformative tools in biomedical research, enabling robust differentiation protocols for diverse cell lineages. This breakthrough allows for the mass cultivation of various aged or diseased cell types in vitro, paving the way for the re-establishment of patient-specific tissues and, in some cases, entire organs. A particularly powerful application has been the development of organoids—complex 3D tissue cultures that faithfully recapitulate organ architecture and function, far surpassing the limitations of conventional 2D cultures [149]. When derived from patient-specific iPSCs, these organoids can precisely mirror individual disease pathologies, as demonstrated by intestinal organoids modeling inflammatory bowel disease while avoiding potential complications seen in other model systems [149,150]. These advancements not only facilitate mechanism studies but also play a pivotal role in drug discovery and testing processes for anti-aging research.

Here we want to emphasize that the process of generating iPSCs involves a phenomenon called age reprogramming, which resets the biological age of cells to a younger state. This rejuvenation effect is evident in various cellular characteristics, including gene expression patterns, and epigenetic markers [151,152]. This age reversal is a key characteristic of iPSC technology and has significant implications for both basic research and potential therapeutic applications. While it allows for the study of early developmental processes and the generation of unlimited cell sources, it can also complicate the modeling of late-onset diseases, especially when a time-dependent aging phenotype is sought, as the rejuvenated cells may not accurately reflect the aged state in which these diseases typically manifest.

Importantly, after prolonged culture (more than one year of continuous culture), iPSCs can be regarded as an aging cell line, exhibiting biological differences in mitochondrial status and nuclear envelope integrity, such as elevated levels of emerin and nesprin-2 and reduced levels of laminin B1 [153]. This may disprove the previous belief that iPSCs can be maintained and propagated indefinitely in culture and retain the ability to re-differentiate into fully rejuvenated cells and thus prolonged culture of iPSCs (21% oxygen) may be useful for studying the aging process and modeling late-onset pathology. However, it has also been shown that both cyclin-dependent kinase inhibitor and SASP genes are prematurely induced in WS fibroblasts, yet the expression levels of these genes are completely suppressed in iPSCs obtained from WS fibroblasts to the same extent as observed in normal iPSCs [154].

Actually, iPSCs have emerged as a powerful tool for investigating aging diseases due to their ability to capture patient-specific genetic backgrounds and recapitulate disease phenotypes. Additionally, iPSCs enable longitudinal studies where cells can be cultured over extended periods, mimicking the gradual nature of aging and disease progression. This approach helps in identifying early markers of disease onset and progression. Here, we provide an overview of how iPSCs-based models are used in different aging-related diseases (Figure 6).

### 4.1. Modeling Human Premature Aging Syndromes with iPSCs

Premature aging syndromes, also termed progeroid syndromes, including HGPS, WS, and CS, encompass a group of rare genetic conditions characterized by accelerated aging and the premature onset of aging-related health issues [155,156,157,158]. The advent of iPSC technology has revolutionized the study of these complex conditions by enabling the in vitro modeling of disease-specific cellular phenotypes [159,160]. This approach provides unprecedented opportunities to investigate molecular mechanisms and develop potential therapeutic interventions.

Among these syndromes, HGPS has emerged as a primary research focus due to its severe nuclear lamina pathology. In 2011, Zhang, J.; et al. induced fibroblasts derived from HGPS patients into iPSCs, and they continued to differentiate into various cells at a slower rate compared to wild type iPSCs [161]. These differentiated cells exhibited elevated levels of progerin, increased DNA damage, and nuclear abnormalities, with VSMC (vascular smooth muscle cells) accumulating a large number of calmodulin-stained inclusion bodies [161]. Similarly, Liu, G.H.; et al. also found that the directed differentiation of HGPS-iPSC into smooth muscle cells leads to the development of a premature aging phenotype associated with vascular aging [162]. Their study also identified the DNA-dependent protein kinase catalytic subunit (DNAPKcs, also known as PRKDC) as a downstream target of progerin, and that the deletion of the nuclear DNAPK holoenzyme is associated with premature aging and physiological aging [162]. This provides a powerful and new tool for unraveling the molecular and physiological mechanisms of premature and normal aging. For example, the potential for high-throughput drug screening using HGPS iPSC-derived cells has been investigated to find compounds for the treatment of HGPS and other aging-related diseases [163,164,165]. Blondel, S.; et al. and Kubben, N.; et al. screened chemical libraries to identify compounds that inhibit progerin processing or ameliorate its pathological effects [163,164]. Lo Cicero, A.; et al. worked on detecting alkaline phosphatase activity in HGPS iPSC-derived MSCs during osteogenic differentiation, and identified seven modulators of premature osteogenic differentiation through a high-throughput drug screen, with all-trans RA (also known as retinoic acid) showing potential therapeutic use for hundreds of other diseases caused by other mutations in the LMNA gene [165].

In WS, specific pure truncation mutant WS fibroblast cell lines are difficult to reprogram using Yamanaka factors because they are defective in inducing hTERT (human telomerase reverse transcriptase), the catalytic unit of telomerase. Therefore, one study restored the ability of this WS fibroblast cell line to form iPSCs through the ectopic expression of hTERT [166].

In addition, Durczak, P.M.; et al. generated integration-free iPSCs from fibroblasts of a CS patient with a mutation in the CSB/ERCC6 gene [150]. They utilized the CRISPR/Cas9 system to correct the mutation, and further obtained homozygous gene-corrected CS-iPSCs. These cells successfully recapitulated the CS-related phenotypic defects in CS-iPSC-derived MSCs and neural stem cells, and these premature aging defects, such as an increased susceptibility to DNA damage stress, can be rescued by targeted correction of the mutated ERCC6 gene [150].

### 4.2. Modeling Human Telomere Dysfunction Disease with iPSCs

Modeling human telomere dysfunction diseases with iPSCs offers a powerful approach to studying the underlying mechanisms and developing potential therapies for these conditions. Telomere dysfunction is associated with various diseases, including DC [167,168], BMF like AA [169,170], and pulmonary fibrosis [41,42,171]. Using iPSCs derived from patients with telomere dysfunction allows researchers to recapitulate disease-specific cellular phenotypes in vitro [172]. The resulting cellular models faithfully reproduce key pathological features observed clinically, including critically shortened telomeres, reduced proliferative capacity, premature senescence onset, and telomere-associated genomic instability [173,174].

DC arises from mutations in six critical genes—five encoding telomerase components (DKC1, TERC, TERT, NOP10, NHP2) and TINF2, which encodes a key telomere-protective protein [175]. Beyond the classic triad of nail dystrophy, abnormal skin pigmentation, and oral leukoplakia, DC can manifest in multiple organs, reflecting the widespread impact of telomere dysfunction on rapidly dividing cells. To investigate these mechanisms, Fernandez et al. established an innovative model using iPSC-derived type II alveolar epithelial cells to study telomere-related pulmonary pathology [176]. Their DC patient-derived cells recapitulated disease hallmarks including telomere shortening, end uncapping, impaired alveolar organoid growth, cellular senescence, and disrupted Wnt signaling. Notably, treatment with the GSK-3 inhibitor CHIR99021 restored Wnt pathway activity, enhanced telomerase function, and rescued the observed phenotypic defects [176].

AA is a life-threatening BMF disease characterized by profound hematopoietic stem cell depletion. Polychronopoulou et al. demonstrated that AA patients exhibit significant telomere shortening in bone marrow mononuclear cells compared to healthy individuals [177]. This finding was extended by Ball et al., who reported that approximately 30% of acquired AA cases show critically shortened telomeres, which correlate with poor clinical outcomes including disease progression, relapse, and reduced survival [178]. iPSCs derived from AA patients can mimic two key features of the disease: (1) the inability to maintain telomere length during reprogramming and hematopoietic differentiation, resulting in shorter telomeres in AA-iPSCs and iPSC-derived hematopoietic progenitors than in controls; and (2) an impaired ability of AA-iPSC-derived hematopoietic progenitors to produce erythroid and myeloid cells [179]. Similar approaches have been applied to Fanconi anemia, the most common types of BMF diseases. Researchers generated integration-free Fanconi anemia iPSCs without genetic correction to avoid insertional mutagenesis risks, successfully recapitulating disease phenotypes while establishing a platform for drug discovery [180]. This model has already identified compounds that improve hematopoietic function, validating its utility for therapeutic development.

Chronic telomere dysfunction specifically in type II AECs—but not in collagen-producing cells—drives age-dependent pulmonary remodeling and fibrotic progression [41]. Specifically, it plays a critical role in the pathogenesis of IPF, an aging-related, progressive lung disease characterized by the excessive accumulation of extracellular matrix within the interstitial layer of the lung parenchyma [181]. iPSCs can also be differentiated into AECs with specific mutations or pro-fibrotic stimuli, which are crucial for understanding the epithelial reprogramming seen in IPF, to establish iPSC-derived lung-like organs that recapitulate the complexity and function of fibrotic lung tissue for disease modeling and drug screening [182]. RNA sequencing analysis revealed that fibrotic conditions induce an aberrant KRT−/KRT17+ basaloid cell population in these models—a pathological cell state also observed in IPF patient lungs [182].

By studying iPSC-derived cell models, researchers can elucidate the molecular pathways involved in telomere maintenance, DNA damage response, cellular senescence, and disease progression [183,184]. This deeper understanding can lead to the identification of novel therapeutic targets and the development of targeted interventions to restore telomere function, enhance cellular proliferation, and alleviate disease symptoms.

The patient-specific nature of iPSC technology revolutionizes drug discovery by permitting pharmacological screening in genetically relevant systems [185]. This personalized medicine paradigm holds particular promise for telomere disorders, where it may accelerate the development of customized therapies that significantly improve clinical outcomes and patient wellbeing.

### 4.3. iPSCs-Based Models for Aging-Related Degenerative Diseases

Traditional animal models often fail to replicate all aspects of human neurodegenerative diseases. iPSC-derived neuronal cultures offer a closer approximation of human brain tissue, thereby improving the translatability of preclinical findings. These models, derived from induced iPSCs, accurately mimic the pathology of aging-related degenerative diseases, offering profound insights into their underlying mechanisms. Below is a refined summary and description of iPSCs-based models for several such diseases.

#### 4.3.1. Alzheimer’s Disease (AD)

iPSCs obtained from patients with AD can be differentiated into neurons, providing researchers with a platform to study disease-specific cellular changes such as amyloid-beta accumulation and tau pathology [186,187,188]. In the study by Chang, K.H.; et al., iPSCs were generated from familial AD patients with Amyloid precursor protein gene mutations exhibiting hallmark pathological changes—elevated Aβ42 levels, reduced synaptophysin expression, and increased caspase 1 activation—all characteristic of AD neurodegeneration [189]. To overcome the challenge of modeling age-dependent tau pathology, Bassil et al. developed an automated culture system that maintains iPSC-derived neurons, astrocytes, and microglia long-term, creating a comprehensive human cell model of AD [190]. This advanced platform enables the observation of tau hyperphosphorylation and synaptic loss that typically only manifest in aged neurons.

CRISPR/Cas9 gene editing has further enhanced AD modeling by allowing the precise introduction of disease-associated mutations. Apolipoprotein E4 iPSC-derived variants are thought to be the greatest genetic risk factor for sporadic AD, and the investigators used CRISPR/Cas9 to create homozygous iPSC lines containing a pure Apo-E4 allele from unaffected parental Apo-E3 cells. They found that Apo-E4 iPSC-derived neurons, astrocytes, and microglia-like cells recapitulated AD-related phenotypes at multiple levels, and the conversion of Apo-E4 to Apo-E3 in brain cell types derived from sporadic AD iPSCs was sufficient to mitigate multiple Alzheimer’s disease-related pathologies [191].

Indeed, early cellular models using human iPSCs for AD and tau proteinopathies only recapitulated the early stages of the disease, and had difficulties in showing protein aggregation, which may be related to the fact that the reprogramming process of iPSCs causes it to go into a “young” state, as mentioned before. And unlike human mES-derived neurons, Sally Martin et al. found that mouse ESC-derived neurons were able to recapitulate the unique DNA methylation profiles of adult neurons within a controlled timeframe of in vitro experiments, which allows them to serve as a model system for studying epigenome maturation during development [192]. One potential issue is that previous models often focused on single cell types, such as neurons, but the brain is a complex organ with multiple interacting cell types [193]. To address this, researchers have developed advanced co-culture systems, automated platforms, and sophisticated 3D models that better capture the brain’s cellular complexity. Shimada et al. made a breakthrough by generating FBOs from feeder-free iPSCs through the precise modulation of fibroblast growth factor 2 concentrations. They established an innovative Adeno-associated virus-mediated gene delivery system for these organoids, successfully creating a tauopathy model that develops tau protofibrils—a valuable tool for understanding tau pathology and developing targeted therapies [194]. Lomoio, S.; et al. combined porous scaffolds made of silk fibroin with collagen hydrogels to prepare a 3D bioengineered model of neural tissue that supported the growth of neurons and glial cells into complex networks [195]. This bioengineered system supported the development of complex neuron–glia networks and, remarkably, recapitulated time-dependent AD phenotypes and patient-specific transcriptomic signatures over 4.5 months of maturation [195].

These models are instrumental in unraveling the molecular mechanisms driving AD, including synaptic dysfunction, neuroinflammation, and oxidative stress [196,197,198,199]. Notably, iPSC-derived human neurons showed strong sensitivity to the synaptotoxic effects of Aβ, which is a feature of AD. aβ specifically damages clusters of axon vesicles, and the amplitude of miniature excitatory postsynaptic currents mediated by aminomethylphosphonic acid receptors is reduced as a result of Aβ exposure, suggesting damage to the postsynaptic aminomethylphosphonic acid receptors [196]. A further mechanistic study by Silva, M.C.; et al. using patient-specific iPSCs revealed that neurons carrying the tau-A152T variant display abnormal tau protein dynamics, including reduced solubility and pathological redistribution [196]. This tau dysregulation correlated with elevated cellular stress markers and heightened susceptibility to multiple stressors, including proteotoxic, excitotoxic, and mitochondrial insults [197]. Importantly, the CRISPR/Cas9-mediated pharmacological activation of tau targeting or autophagy could attenuate these vulnerabilities, thus suggesting potential therapeutic approaches, highlighting promising therapeutic avenues [197].

Neuroinflammatory responses in AD have been effectively modeled using iPSC-derived glial cells. AD patient-derived astrocytes demonstrate the elevated baseline expression of GFAP and S100β, along with enhanced Aβ42 secretion and phagocytic activity [198]. Similarly, AD microglia exhibit impaired IL-8 production under resting conditions, but mount exaggerated pro-inflammatory responses when activated, releasing IL-18 and MIF, while astrocytes overproduce IL-6, CXCL1, ICAM-1, and IL-8 [198].

Mitochondrial dysfunction represents another critical aspect of the AD pathology that is closely tied to the oxidative stress mechanisms. Lee, S.E.; et al. verified that the accumulation of amyloid precursor protein C-terminal fragments resulted in mitochondrial phagocytosis failure by inducing neural stem cells derived from AD patients [200]. These include abnormal LC3-II and p62/SQSTM1 accumulation, impaired PINK1/Parkin recruitment to mitochondria, and failed mitochondrial–lysosomal fusion [200]. Such insights are crucial for advancing drug discovery efforts and the development of therapeutic interventions.

#### 4.3.2. Parkinson’s Disease (PD)

iPSC-derived dopaminergic neurons from PD patients exhibit key disease features such as alpha-synuclein aggregation and impaired dopamine neurotransmission. Vuidel et al. employed an innovative automated imaging platform combining high-content analysis with machine learning to systematically characterize PD phenotypes in patient-derived midbrain dopaminergic neurons [201]. Their approach successfully distinguished genetic subtypes by detecting pathological markers like elevated α-syn levels, increased Ser129 phosphorylation, simplified neuronal arborization, and mitochondrial impairment, while simultaneously screening LRRK2 and α-syn-targeting therapeutics [201]. Interestingly, Kim, M.S.; et al. developed and applied an optogenetically assisted α-syn aggregation induction system. This system was designed to rapidly induce α-syn aggregation and toxicity in a PD model using human iPSCs-derived midbrain dopaminergic neurons and midbrain organoids, thereby facilitating the study of pathologies related to PD and the screening of potential therapeutic compounds [202]. This powerful platform not only accelerates PD research, but also enabled the identification of promising drug candidates like BAG956, which demonstrated therapeutic potential by enhancing the autophagic clearance of pathological α-syn aggregates [202].

Researchers have developed an innovative model of PD pathology by treating iPSC-derived dopaminergic neurons with preformed α-synuclein fibrils to induce Lewy body-like inclusions [203]. To better mimic the inflammatory microenvironment of PD, this approach incorporates immune challenges through either pro-inflammatory cytokine treatment (interferon-γ or interleukin-1β) or co-culturing with activated microglia [203]. This combined model has allowed researchers to study the effects of immune dysfunction and lysosomal damage on the formation of LB-like inclusion bodies, providing insight into the pathomechanisms of PD. Moreover, Gustavsson, N.; et al. advanced PD modeling by integrating iPSC technology with advanced biophysical techniques. Their comparative analysis of patient-derived versus control neurons using vibrational spectroscopy revealed that PD cellular environments promote distinct structural modifications in α-syn aggregates [204]. These findings suggest that genome-induced changes in the environment of patient cells could trigger structural changes in α-syn, which may lead to the formation of strains with different structures, properties, and seeding tendencies [204].

These models help researchers delve into the pathogenic pathways underlying PD, including mitochondrial dysfunction, protein aggregation, and neuroinflammation [201,202]. While many studies have shown general disease features, there is room for more detailed investigation into how specific genetic mutations (e.g., SNCA, LRRK2) influence these phenotypes [205]. The study by Chedid, Jessica et al., using PD patient-derived iPSCs from different genetic mutations, found that (1) parkin RBR E3 ubiquitin protein ligase loss-of-function mutations affect lysosomal and mitochondrial function in dopaminergic neurons, leading to an early selective loss of dopaminergic neurons; (2) SNCA A53T mutated cells exhibit more subtle cellular abnormalities such as mitochondrial and autophagy dysfunction, which over time may lead to the aggregation of α-synuclein and the deposition of tau protein; (3) the LRRK2 R1441G mutation may also lead to tau protein deposition in neurons, but the pathological accumulation of α-syn was not evident, exhibiting reduced glucosinolase activity and increased phosphorylation of α-syn [205]. These findings contribute to the understanding of how lysosomal and mitochondrial dysfunction differ in different PD subtypes and how they affect the pathogenesis of the disease.

Neuroinflammation, a recognized driver of PD, coexists with intriguing neuroprotective adaptations during disease progression. Gerasimova, T.; et al., employing iPSC-derived neuronal models, uncovered a multidimensional self-defense system activated by PD neurons to counteract pathological insults. Notably, neurons with genetic vulnerabilities characteristic of PD demonstrate the adaptive reprogramming of gene expression, marked by the coordinated suppression of pro-inflammatory pathways and apoptotic signaling. Further investigations have revealed that stressed neurons engage receptor-mediated signaling networks to restore homeostasis. For instance, glia-derived cytokines modulate stress-responsive receptors, while neurons autonomously upregulate the synthesis and secretion of apelin—a multifunctional protective polypeptide. These coordinated responses highlight a compensatory mechanism wherein PD neurons reinforce survival pathways through both paracrine and autocrine regulation [206].

Such investigations are pivotal in the quest for novel treatments and neuroprotective strategies for PD. However, longitudinal studies for PD could provide insights into the progression of these features over time, which would be more valuable for understanding disease dynamics. Future research should focus on refining these models to better reflect the dynamic nature of PD over time.

#### 4.3.3. Huntington’s Disease (HD)

HD is a neurodegenerative late-onset genetic disorder caused by a CAG amplification in the coding region of the Huntingtin gene, resulting in toxic polyglutamine stretches in the huntingtin protein. Back in 2010, Zhang, N.; et al. used iPSCs to mimic the cellular phenotype of HD [207]. The iPSCs were established from patients with low CAG repeat expansion in the Huntingtin gene, which was then efficiently differentiated into neurons. Subsequent studies have established iPSC-derived neurons as robust HD models that recapitulate key pathological hallmarks, including protein aggregation, neuronal dysfunction, and selective cell death. For example, as Nekrasov, E.D.; et al. reported, the generated HD GABAergic multiple sclerosis-like neurons display mutant Huntingtin aggregates, abnormal lysosomal/autophagosomal proliferation, nuclear deformations, and age-dependent neurodegeneration [208]. Interestingly, the ectopic expression of Progerin in wild-type and HD-derived iPSC neurons exacerbated the otherwise insignificant changes in gene expression between these cells, suggesting that the IGF1 and genes involved in neurogenesis and neurologic development are persistently altered in Huntington’s disease cells [209]. Indeed, there have also been studies in which neural progenitor cells were differentiated from iPSCs of transgenic HD monkeys into mature neural cells in vitro, such as neurons and glial cells [210]. A recent study by Wu, G.H.; et al. utilized cryo-electron tomography to observe iPSC-derived neurons (with distinct CAG repeat sequences) from HD patients. Besides this, ultrastructural and proteomic approaches were integrated to discover possible early HD phenotypes. For example, lamellar aggregates with distorted cristae and enlarged mitochondrial RNA granules were found in double membrane organelles and mitochondria in HD iPSCs [211]. This knowledge opens doors for targeted therapeutic interventions aimed at mitigating the devastating effects of HD.

#### 4.3.4. Amyotrophic Lateral Sclerosis (ALS)

iPSC-derived motor neurons derived from ALS patients showcase disease-specific characteristics like TDP-43 pathology, mitochondrial abnormalities, and motor neuron degeneration [212,213,214].

One of the most prominent features of ALS is the cytoplasmic mislocalization, aggregation, and phosphorylation of TDP-43 [215]. Using iPSCs derived from ALS patients, Ritsma, Laila et al. discovered that the aggregation of TDP-43 is often due to an impaired proteasomal degradation process. This impairment leads to the mis-splicing of the neuronal growth-associated factor stathmin 2 (STMN-2), resulting in axonal degeneration [215]. In a further study, Lépine, S.; et al. utilized CRISPR/Cas9-engineered iPSC strains carrying the TDP-43 mutation, and found different results. Interestingly, mutant motoneurons did not exhibit typical ALS pathological features associated with TDP-43, such as obvious aggregation, increased phosphorylation, and abnormal nuclear-cytoplasmic distribution. Instead, they showed a gradual weakening of spontaneous neural activity, characterized by reduced electrophysiological function, and abnormalities in synaptic structure and function [212]. These models serve as essential tools for investigating ALS pathophysiology, including defects in RNA metabolism, neuroinflammation, and glutamate excitotoxicity [214,216].

The C9ORF72 mutation represents the most common genetic cause of ALS, exhibiting autosomal dominant inheritance. Li et al. employed a multi-omics approach (genomics, transcriptomics, epigenomics, and proteomics) to systematically characterize molecular alterations in iPSC-derived motor neurons from C9ORF72 ALS patients [216]. By cross-validating these findings in a C9ORF72 Drosophila model, the team successfully identified both pathogenic mechanisms and compensatory pathways [216]. These first attempts to link the many different aspects of cellular function are a starting point for further research, and provide a paradigm for developing a holistic view of cellular function in the face of potentially pathogenic repetitions of ALS.

Mitochondrial dysfunction represents another key pathological feature of ALS frequently co-occurring with TDP-43 proteinopathy. iPSC-derived motor neurons recapitulate the mitochondrial defects observed in ALS patients, particularly inner membrane structural abnormalities [217]. Notably, the small molecule SBT-272 demonstrates therapeutic potential by stabilizing cardiolipin in the mitochondrial inner membrane, which improves mitochondrial structure, motility, and function in TDP-43-affected upper motor neurons [217].

Additionally, the traditional focus on motor neuron-centered mechanisms has recently shifted to understanding the role of non-neuronal cells (e.g., microglia) in the pathophysiology of ALS [218]. For example, the effect of the C9ORF72 mutation was investigated using iPSC-derived microglia to study phagocytosis and immune responses under lipopolysaccharide stimulation [219]. To investigate the effect of autophagy defects on neuronal vulnerability, co-culture studies with motor neurons were also performed, and these revealed that autophagy defects in C9ORF72 knockout iPSC-MG increased the vulnerability of motor neurons to excitotoxic stimuli, suggesting the dysfunction of cellular autoimmunity, and that the pharmacological activation of autophagy ameliorated microglial cell functional defects and motor neuron death [219]. These findings reveal an important role for C9ORF72 in the regulation of immune homeostasis, and identify the dysregulation of the myeloid lineage as a factor contributing to neurodegeneration in ALS.

Recent advances have further characterized the neurotoxic transformation of mutant microglia, which exacerbate neuroinflammation and accelerate motor neuron degeneration [218]. In addition, as in other neurodegenerative diseases, Guo et al. successfully generated spinal cord organoids derived from C9ORF72 knockout human iPSCs to mimic ALS disease and screen for unrevealed phenotypes [220]. The differentiated models showed the low expression of C9ORF72 and exhibited pathological features of ALS, especially neuroinflammation [220]. This highlights the importance of iPSC models in understanding the role of neuroinflammation in ALS progression. However, there is still room for improvement, particularly in refining 3D culture systems and integrating multiple cell types to better mimic the complexity of the human nervous system.

These works have allowed us to look more closely at the impacts of mutations in different cells in the brain on the development of ALS. Such investigations pave the way for potential ALS therapies and disease-modifying interventions.

#### 4.3.5. Age-Related Macular Degeneration (AMD)

iPSC-derived models have successfully recapitulated key features of AMD, including drusen formation, RPE atrophy, mitochondrial dysfunction, and genetic polymorphisms in the complement pathway [221]. Notably, these models have revealed that AMD high-risk alleles (CFH and ARMS2) disrupt RPE pigmentation by activating NF-κB signaling while suppressing autophagy pathways [221]. Through high-throughput drug screening, Sharma et al. identified L-745,870 (a dopamine antagonist) and aminocaproic acid (a protease inhibitor) as compounds that normalize RPE pigmentation and epithelial morphology even in CFH(H/H) genotype cells [221].

Mitochondrial impairment represents a central feature of AMD pathogenesis, including decreased mitochondrial mass and electron transport chain protein content, increased mtDNA damage, and reduced mitochondrial function [222]. A recent study by Mara, C. Ebeling et al. showed that iPSC-RPE cells with the CFH high-risk allele, one of the most prevalent AMD-associated single-nucleotide polymorphisms, exhibited a more pronounced reduction in mitochondrial function and increased inflammatory markers, regardless of the presence of AMD [223]. Additionally, Bhattacharya et al. introduced a specific mitochondrial defect into iPSC-RPE cells, and found that when the mitochondria in these cells do not work properly, they disrupt the cell’s energy supply and the process of cleaning up damaged mitochondria (known as mitochondrial phagocytosis) [224]. This disruption leads to damage to RPE cells, which can lead to aging-related macular degeneration.

These applications employ the iPSC-derived model as a valuable tool for understanding the underlying mechanisms of AMD and developing potential therapeutic strategies. However, the further refinement of these models is necessary to enhance their predictive power and translational potential. Especially, the current models primarily focus on RPE cells, but interactions with other cell types, such as photoreceptors and immune cells, could provide a more comprehensive understanding of AMD pathogenesis.

### 4.4. Modeling Aging-Related Metabolism Diseases with iPSCs

A critical comparison between iPSCs from young and aged donors reveals significant mitochondrial dysfunction in the latter, characterized by impaired bioenergetics and elevated ROS production [225]. This persistence of aging phenotypes at the mitochondrial level makes aged donor-derived iPSCs particularly valuable for drug screening aimed at improving mitochondrial function. One of the primary reasons for impaired mitochondrial function in aged iPSCs is the accumulation of somatic mtDNA mutations. Kang et al. showed that while pooled skin and blood mtDNA from elderly subjects contain low hetero-plasmic point mutations, individual iPSC lines derived from these tissues carry an elevated load of such mutations [226]. Additionally, iPSCs from aged donors also exhibited excessive glutathione-mediated ROS scavenging activity, which could paradoxically contribute to mitochondrial dysfunction by altering redox homeostasis [227]. These observations have led researchers to delve into understanding the mechanisms behind these phenomena. Here we introduce the common uses of iPSC-derived models for aging-related metabolism diseases.

#### 4.4.1. Type 2 Diabetes Mellitus (T2DM)

Su et al. used iPSC-derived ECs from patients with T2DM to develop in vitro models of diabetic endothelial dysfunction. They observed that the diabetic iPSC-ECs exhibited increased ROS production, decreased nitric oxide bioavailability, and impaired angiogenic capacity compared to control ECs [228]. These findings align with the known pathophysiology of diabetic vascular complications. Advancing beyond single-cell-type models, Tao et al. developed a sophisticated microfluidic platform supporting the 3D co-culture of iPSC-derived liver and pancreatic islet organoids [228]. The organ tissues were cultured under circulating perfusion conditions for the replication of the human liver–pancreas–islet axis, which helped in studying the interactions between hepatocytes and islet cells in normal and disease states [228]. Such an improvement can enhance tissue-specific functionality and allows transcriptional analyses to study metabolic signaling pathways, glucose-stimulated insulin secretion, and glucose utilization.

#### 4.4.2. Diabetic Cardiomyopathy (DCM)

Using human iPSC-CMs, Horikoshi et al. showed that iPSC-CMs matured in medium containing fatty acids but not glucose, and when exposed to 11 mM glucose in a diabetic environment, they exhibited diabetic cardiomyopathy phenotypes, including increased cell size and a significant increase in brain natriuretic peptide secretion [229]. In a similar study by Carter et al., human iPSC-CMs successfully induced insulin resistance and mimicked the metabolic features and diastolic dysfunction of T2DM after being cultured for 6 days in medium with or without insulin, respectively [230]. In particular, using a 3D culture of human iPSC-CMs as engineered heart tissue activated multiple pathways associated with diabetic cardiomyopathy, including metabolic remodeling, mitochondrial dysfunction, extracellular matrix remodeling, and increased endoplasmic reticulum stress [230].

#### 4.4.3. Nonalcoholic Fatty Liver Disease (NAFLD)

iPSC-derived hepatocytes from NAFLD patients displayed several metabolic abnormalities indicative of NAFLD, such as increased lipid accumulation and altered mitochondrial function, and inflammatory response [231], even in the absence of any metabolic challenge. The iPSC-derived liver organ-on-a-chip system was developed to expose liver organoids to free fatty acids in perfused 3D cultures, thus providing insights into the potential mechanisms underlying steatosis and mimicking the pathological features of NAFLD [232].

### 4.5. Modeling Premature Ovarian Aging with iPSC

Normally, the ovaries stop functioning and begin to decline earlier than other organs in the body, thus initiating systemic aging. Ovarian aging causes hormonal, tissue and cellular changes, as well as systemic effects on other organs, leading to a range of negative health consequences that ultimately affect well-being and personal freedom [233]. POI refers to the condition in which women experience a loss of ovarian function at an age earlier than the estimated average age of menopause. Due to the complexity and specificity of POI, there are currently no specific treatment methods available for this condition.

Recent advances in reproductive biology have resulted in significant progress in germ cell generation. Yamashiro et al. developed a comprehensive program that demonstrated the successful differentiation of iPSCs into functional oocytes through germ cell lineage specification [234,235]. Subsequently, Murase Y.; et al. found that human primordial germ cell-like cells derived from human iPSC could proliferate at least approximately 106-fold within 4 months under specific in vitro conditions [236]. This breakthrough highlights the complex developmental trajectory of mammalian germ cells, which involves a series of precisely regulated cellular events and fate transitions, and also implies that it is possible to generate models of ovarian aging using iPSCs (Figure 7). Indeed, ovary-like organogenesis protocols using stem cell-derived cells have recently been available, but there is still a void in the iPSC field [237,238].

In summary, iPSCs have revolutionized the investigation of aging and aging-related diseases by enabling the study of patient-specific pathologies in vitro. However, ongoing improvements in modeling techniques will be crucial for realizing their full potential in both research and therapeutic applications. iPSCs can be differentiated into various cell types affected by premature aging syndromes, telomere dysfunction diseases, and aging-related degenerative diseases, such as fibroblasts, endothelial cells, and hematopoietic cells. This allows researchers to create disease models that closely mimic the in vivo environment. It should be noted that while iPSC models offer many advantages, they may not fully capture all aspects of complex diseases. For example, some models might lack the microenvironmental cues present in vivo, potentially affecting the accuracy of the results. Additionally, ensuring consistent reprogramming efficiency and maintaining genomic stability over time remain challenges.

## 5. Induced Pluripotent Stem Cell-Derived Differentiated Cells for Aging Diseases Therapy

Aging-related diseases often involve the dysfunction or loss of specific cell types, leading to organ and tissue degeneration. Due to their “young” characteristics, iPSCs offer a promising solution by enabling the reprogramming of adult cells into a pluripotent state, which can then be directed to differentiate into various cell types needed to replace damaged or dysfunctional cells and thus make a difference in aged bodies [120]. Below is a summary and description of iPSC-derived differentiated cells used for aging disease therapy.

### 5.1. Patient-Specific Treatment

The iPSC technology allows for the generation of patient-specific differentiated cells, reducing the risk of immune rejection and improving treatment efficacy [120,240,241]. One of the primary ways in which iPSCs help mitigate immune rejection is through autologous transplantation, whereby cells derived from a patient’s own somatic cells are reprogrammed into iPSCs and then differentiated into the desired cell type for therapy. This approach theoretically minimizes immune rejection because the transplanted cells originate from the same individual (Figure 8). However, it has been noted that even autologous iPSC-derived cells can occasionally be rejected by genetically identical recipients, indicating that other factors may influence immune responses [242].

Recent breakthroughs in gene editing have facilitated the development of hypoimmunogenic iPSCs designed to evade host immune responses. By knocking out specific genes involved in immune recognition, such as B2M, which is crucial for major histocompatibility complex class I expression, researchers aim to create cells that are less likely to trigger an immune response. This strategy is linked to creating “universal donor” iPSC lines that can be used off-the-shelf for multiple patients. For instance, deleting adhesion ligands CD54 and CD58 on target cells significantly reduces NK cell reactivity [243]. Similarly, Chimienti, R.; et al. showed that abrogating major histocompatibility complex class I expression can address immune-mediated responses, but might activate NK cells through missing self-recognition mechanisms [244]. Next, Tsuneyoshi et al. developed a low-immunogenic clone of human iPSCs called HyPSCs by knocking out the HLA class Ia and II genes and introducing the exogenous B2M, HLA-G, programmed death-ligand 1, and programmed death-ligand 2 genes [245]. These HyPSCs retained the normal karyotype and pluripotent stem cells, and were capable of differentiating into all three lineages as well as multiple cell types, including CD45+ hematopoietic progenitor cells, functional endothelial cells, and hepatocytes [245]. Importantly, HyPSC-derived HPCs had the ability to evade both innate and adaptive immune responses, and a synthetic molecule, rapamycin-activated caspase 9, serves as its safety switch, as the exogenous rapamycin can induce caspase 9 activation [245]. Such universal donor cells could provide a scalable solution for cell therapy without the need for personalized cell production. All of the above advances represent a major step toward scalable, off-the-shelf cell therapies that could transform regenerative medicine by overcoming immune rejection barriers.

This personalized approach is particularly advantageous in aging diseases where individual variations play a significant role in disease progression and response to therapy.

### 5.2. Cell Replacement and Regeneration

Cell therapy has recently emerged as a promising approach to repair or replace damaged tissue, as well as engineer immune responses to a disease, such as cancer. iPSC-derived differentiated cells have the potential to replace damaged or senescent cells in aging tissues and organs (Figure 9).

For example, iPSC-CMs can be used to regenerate heart tissue in cardiovascular diseases [246,247], while iPSC-derived neurons can be utilized for neurodegenerative disorders such as PD [248,249]. The primary advantage of using iPSC-CMs is that they do not pose significant ethical issues and suffer negligible immunological rejection compared to other myocardial regeneration methods. Innovative engineering approaches, such as Chang et al.’s CRISPR-edited iPSC line expressing cyclin-D2 and light-sensitive ion channels, now enable precise control over transplanted cardiomyocyte activity [250]. Several studies have explored the potential use of iPSC-CMs in regenerating heart tissue. For instance, large animal studies have demonstrated that the intramyocardial injection of human iPSC-CMs following myocardial infarction results in cell grafts, although there are concerns about ventricular arrhythmias [251]. Another study by Fang et al. focused on generating hypoimmunogenic hiPSCs to overcome the issue of post-transplant rejection due to HLA mismatching [252]. To improve the survival of implanted cells and reduce the risk of arrhythmogenesis, Cheng et al. also generated human iPSC-derived ECs and human iPSC-CMs to enhance vascularization and promote the maturation of the transplanted cells [253]. Additionally, 3D stem cell-derived cardiac models have been developed to better mimic the intricate microenvironment present in the heart [254]. Despite this progress, key challenges must be resolved before clinical translation, including optimizing cell survival, ensuring robust vascular integration, and eliminating arrhythmia risks [255]. Furthermore, developing more sophisticated models that fully capture human cardiac tissue complexity remains critical for advancing the field [256].

Recent advances in iPSC-based therapies demonstrate remarkable potential for Parkinson’s disease treatment across multiple model systems. As a study by Guo et al. showed, the iPSCs derived from minipigs were also differentiated into GABA progenitor cells and transplanted into the right medial forebrain bundle of PD model rats. The transplanted cells survived and differentiated into various types of neurons, including GABAergic and dopaminergic neurons, as well as glial cells, which were transplanted into the surrounding brain tissues, including the striatum and substantia nigra, for at least 32 weeks, and this ultimately improved the functional recovery of the PD rats, helping them to overcome the behavioral deficits of PD [248]. Translating these findings to human applications, researchers first validated iPSC-derived midbrain dopaminergic progenitor cells in humanized mouse models before progressing to clinical trials [257]. Fortunately, these cells were sequentially transplanted into a patient’s putamen (left and right hemispheres six months apart), resulting in stabilized or improved PD symptoms at 18-24 months post-implantation [257]. Notably, human midbrain organoid tissues consisting mainly of midbrain dopaminergic cells obtained by iPSC differentiation reversed motor dysfunction and established bi-directional connectivity with natural target areas of the brain without tumor formation or graft overgrowth after transplantation into the striatum of a mouse model of PD after 15 days in culture for 12 weeks [249].

In addition, iPSCs differentiated into TM cells, termed iPSC-TM, can regulate intraocular pressure and prevent neuronal loss in a glaucoma mouse model, thus restoring TM function for over 9 weeks [258]. Interestingly, by optimizing the type and number of transcription factor-containing vectors, as well as the transfection load and culture medium, Tanaka, N.; et al. improved the efficiency of inducing iPSCs from skin fibroblasts of elderly patients, and successfully induced iPSCs from myocardial fibroblasts isolated from the pathological heart tissue of heart transplant recipients [259]. This demonstrated that the improved reprogramming method can be used to generate iPSCs from a variety of pathological and aging tissues, thus advancing the application of iPSC for aging-related diseases and the treatment of cellular senescence [259].

Interestingly, if human follicular fluid is added to the stem cell differentiation system, it significantly enhances the steroidogenic potential of the differentiated stem cells. Elias et al. demonstrated this potential by transplanting iPSC-derived stem cells into the ovaries of chemotherapy-treated infertile mice, successfully restoring ovarian hormone production and generating functional oocytes [260]. This achievement suggests that these cells are capable of replacing damaged ovarian tissue and re-establishing normal hormone secretion. However, significant challenges remain before clinical application can be considered. The resulting oocytes must undergo rigorous validation to ensure they are genetically stable, epigenetically normal, and fully competent for both fertilization and supporting healthy embryonic development. Overcoming these hurdles will be critical for translating this promising technology into safe and effective fertility treatments.

As previously discussed, iPSC-based cell therapy can also fall into two categories—autologous and allogeneic—and both are being tested via clinical trials. In fact, iPSC is currently being used in far more therapeutic settings than just the above examples, and is widely distributed in diseases of various organs [120]. Of the ongoing clinical trials, one using allogeneic iPSC-derived dopaminergic progenitor cells for the treatment of PD has progressed most rapidly to a Phase III clinical trial. Other trials are almost exclusively between Phase I and Phase II, such as those on using autologous iPSC-derived RPE for macular degeneration, and allogeneic iPSC-derived neural/endothelial progenitor cells for ischemic stroke [120]. This clinical landscape demonstrates the rapidly expanding potential of iPSC technology across multiple therapeutic areas.

### 5.3. Tissue Engineering and Organ Transplantation

iPSC-derived cells are integral to tissue engineering approaches for creating functional tissues and organs. With the help of other tissue engineering structures such as hydrogels, engineered tissues can be smoothly molded and even transplanted into patients to restore organ function and alleviate symptoms associated with aging diseases, such as osteoarthritis or liver degeneration (Figure 10).

iPSC-derived chondrocytes encapsulated in gelatin methacryloyl hydrogels produce stable 3D constructs of hyaline cartilage [261]. After 21 days of in vitro culture, iPSC-derived chondrocytes encapsulated in gelatin methacryloyl hydrogels maintained their chondrocyte phenotype in regular medium, and retained their matrix-forming capacity in vivo [258]. This technique is attractive for generating both immature and more mature hyaline cartilage-like tissues, which is significant because the implantation maturity required for articular surface regeneration is unknown.

The field of liver regeneration achieved a major breakthrough in 2013 when Takebe et al. demonstrated that transplantable, vascularized human liver tissue could be generated from iPSCs. Their innovative approach involved creating iPSC-derived liver buds—three-dimensional structures wherein hepatocytes self-organized and formed functional vascular connections with host blood vessels, enabling maturation into adult-like liver tissue [262]. Currently, in order to increase the efficiency of model construction, studies have been conducted to establish integrated differentiation platforms that are scalable and free of xenogeneic components to generate hepatic progenitor cells, mesenchymal stromal cells and endothelial cells. For example, Saeed, Abbasalizadeh et al. generated cell-carrying microcapsules using a scalable microfluidic system with a hydrogel as the shell material [263]. Recently, researchers like Kojima et al. and Minami et al. have used liver decellularization to create acellular scaffolds that can be repopulated with iPSC-derived hepatic cells [264,265]. Kojima et al. demonstrated that when these iPSC-derived hepatic cells were cultured on these scaffolds under circumfusion conditions, they exhibited superior functionality, including increased albumin secretion and up-regulated cytochrome P450 activities, compared to standard two-dimensional cultures. The iPSC-derived hepatic cells also showed long-term survival and vascularization when implanted into immunodeficient microminiature pigs, indicating the potential of these scaffolds for use in regenerative medicine and research [264].

Furthermore, thanks to bioprinting technology, iPSC-derived cells can be printed into these 3D structures to form functional tissues. This process typically involves the use of bioinks, which are materials that support cell viability and promote tissue formation. Studies have shown that iPSC-derived cells can be embedded within bioinks such as hyaluronic acid or gelatin-methacrylate/fibrin-based matrices, which provide structural support while allowing for cellular interactions and differentiation [266,267]. In addition, scaffold-free methods, such as the Kenzan method for bio-3D printing, have been employed to fabricate cartilage constructs using iPSC-derived cells, addressing some limitations associated with traditional scaffold-based techniques [268]. This approach helps overcome issues like poor cytocompatibility and degradation-associated toxicity. However, there are still challenges to overcome, such as improving tissue maturation and ensuring long-term functionality.

In summary, iPSC-derived cells are critical to tissue engineering because they provide a versatile source of cells capable of forming functional tissues and organs. They also present key advantages, including the avoidance of ethical concerns and the potential for personalized medicine. However, challenges remain regarding the integration of iPSC-derived tissues with host tissues and ensuring long-term survival. Additionally, optimizing differentiation protocols to achieve mature cell phenotypes remains an area requiring further research.

### 5.4. Therapeutic Paracrine Secretion

iPSC-derived differentiated cells can be engineered to secrete beneficial factors that promote tissue repair, reduce inflammation, and enhance cellular function. Many studies have shown that iPSC-derived MSC have plasticity and immunomodulatory capabilities, which may contribute to the unique therapeutic potential of MSC. For example, Human iPSC-derived MSC from old people can express rejuvenation associated factors (e.g., angiopoietin, Dickkopf-1, Interleukin 8, Platelet-Derived Growth Factor AA, osteoblastin, Serpin Family E Member 1 and Vascular endothelial growth factor) [269]. This paracrine signaling mechanism not only overcomes the limitations of aging in adult MSCs, but also highlights their promising use in clinical applications (Figure 11).

In the context of heart regeneration, iPSC-CMs and iPSC-cardiac progenitor cells not only integrate into damaged heart tissue, but also secrete paracrine factors that stimulate endogenous repair mechanisms. These factors can rescue injured cardiomyocytes by modulating apoptotic pathways and restoring the peri-infarct region in myocardial infarction, or repairing hypoxia-injured cardiomyocytes in heart failure [270,271]. Importantly, this paracrine activity enhances neovascularization and reduces fibrosis, both of which are essential for improving cardiac function post-injury.

Similarly, exosomes derived from iPSC-based therapies show broad regenerative potential. For example, exosomes from iPSC-derived MSC have been used to treat osteoarthritis by promoting cartilage regeneration and reducing joint inflammation [272]. By modulating signaling pathways related to aging, these exosomes isolated from iPSCs might rejuvenate aged cells or prevent premature senescence [273]. In ophthalmology, exosomes from iPSC-RPE cells can protect and restore RPE function by delivering therapeutic molecules and enhancing metabolic activity [274], offering a non-invasive method to treat retinal degenerative diseases like AMD [275]. Interestingly, more and more studies are using exosomes derived from different kind of stem cells to rescue ovarian aging in POI models [276,277,278], but there is still a gap in the iPSC field. These exosomes could serve as an optimal delivery system for bioactive molecules due to their safety and efficacy. However, further research is needed to fully characterize their cargo and optimize their therapeutic potential.

### 5.5. iPSC-Induced Immunotherapy for Anti-Aging

In the last few years, the concept of senolytics has been gradually pursued, with CAR-T therapies targeting senescent cells being very innovative [279,280]. Currently, hPSC and iPSC are capable of differentiating various immune cells, and are loaded with human CAR through cell engineering to generate immune cells with specific targeting properties, including CAR-T cells, CAR-neutrophils, CAR-NK cells, and CAR macrophages for immunotherapy [281,282,283,284]. However, there is still a gap in the development of stem cell-induced immunotherapy used for anti-aging. iPSC-based immunotherapies may help counteract immunosenescence by providing a source of young, functional immune cells. Moreover, iPSC-derived antigen-presenting cells, such as dendritic cells, can be engineered to stimulate the patient’s own immune system to target age-related pathologies in the future. Genetic modifications can also be used to improve the safety of iPSC-derived T cells by controlling activation and preventing off-target effects [285]. Thus, iPSC-derived T cells offer a promising platform for “off-the-shelf” immunotherapy, providing a readily available source of T cells that can be manufactured and cryopreserved for future use.

## 6. Conclusions and Future Perspective

In conclusion, the emergence of iPSCs has opened new frontiers in the field of regenerative medicine and aging research. These remarkable cells provide a unique platform for modeling aging-related diseases and understanding the complex mechanisms of aging at a cellular level. Their ability to differentiate into a wide array of cell types offers unprecedented opportunities for developing personalized regenerative therapies that could potentially replace aged or damaged tissues, thereby extending health span and improving quality of life for the aging population.

Despite the promising potential of iPSC technology, several challenges remain to be addressed before its full therapeutic potential can be realized. These include ensuring the safety and stability of iPSC-derived cells, overcoming potential immune rejection issues, and refining differentiation protocols to produce fully functional and mature cell types. Additionally, establishing robust protocols for large-scale production and rigorous quality control will be essential for the successful clinical translation of iPSC-based therapies.

The field of iPSC-based cell therapy is advancing rapidly, with genetic engineering and cellular manipulation techniques significantly enhancing the functionality and therapeutic potential of iPSC-derived cells. As research progresses, the integration of cutting-edge iPSC technology with discoveries in aging biology promises to revolutionize treatments for aging-related diseases. Future advancements in iPSC applications for aging will depend on improving safety measures, developing scalable production protocols, and optimizing delivery methods. Ensuring the safety of iPSC-based therapies is paramount. The potential risks associated with iPSCs include tumorigenicity, immunogenicity, and genetic instability. For instance, when considering iPSC-derived RPE cells for treating AMD, the safety profile of these cells must be rigorously evaluated to prevent adverse reactions. Similarly, the reduced regenerative capacity of aged iPSC-derived cardiomyocytes underscores the need for rigorous quality control in cardiac applications. Scalable protocols for the expansion, differentiation, and purification of iPSCs are necessary to meet clinical demands. For example, the scalable production of iPSC-derived immune cells, such as NK/ILC cells, has been demonstrated in clinical trials for cancer immunotherapy. Additionally, the scalable generation of red blood cells from iPSCs has shown promise, highlighting the feasibility of producing therapeutic cell types in sufficient quantities. Finally, efficient and effective delivery methods are essential for translating iPSC-based therapies into clinical practice. Advanced 3D culture systems and organoids provide better models for testing delivery methods and assessing therapeutic outcomes. Moreover, innovative delivery systems, such as extracellular vesicles, offer advantages in targeted drug delivery, including high biocompatibility and stability.

Collaborative efforts between researchers, clinicians, and industry stakeholders will be essential to accelerate the translation of iPSC-based therapies from bench to bedside. By combining iPSC technology with cutting-edge tools like CRISPR gene editing, advanced bioengineering, and AI-driven analytics, we can significantly improve our capacity to tackle age-related degeneration at its source.

Beyond merely treating aging symptoms, iPSCs offer the transformative potential to intervene in fundamental aging processes, ushering in a new paradigm of regenerative medicine focused on extending both lifespan and healthspan. As these technologies advance, it is crucial to maintain a focus on ethical considerations and regulatory frameworks to ensure that these groundbreaking therapies are developed responsibly and equitably.

## Figures and Tables

**Figure 1 cells-14-00619-f001:**
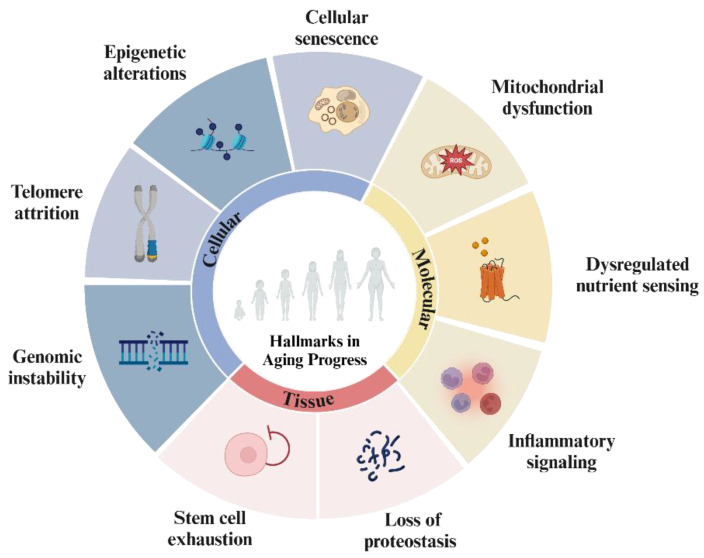
Hallmarks of aging. Tissue, cellular, and molecular mechanisms are interconnected, collectively contributing to the overall aging phenotype. Created with BioRender.com.

**Figure 2 cells-14-00619-f002:**
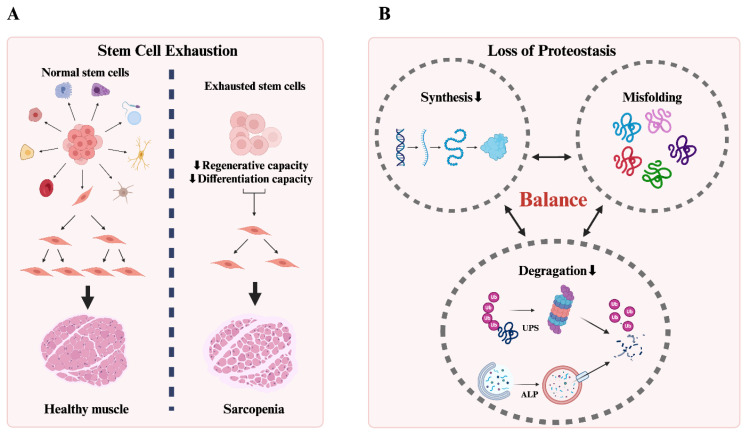
(**A**) Stem cells and their differentiation and paracrine decline lead to Sarcopenia. (**B**) Protein homeostasis consists of a regulatory network that controls protein synthesis, folding, and degradation, and once the balance is disrupted it disrupts the integrity and functionality of the proteome. Created with BioRender.com.

**Figure 3 cells-14-00619-f003:**
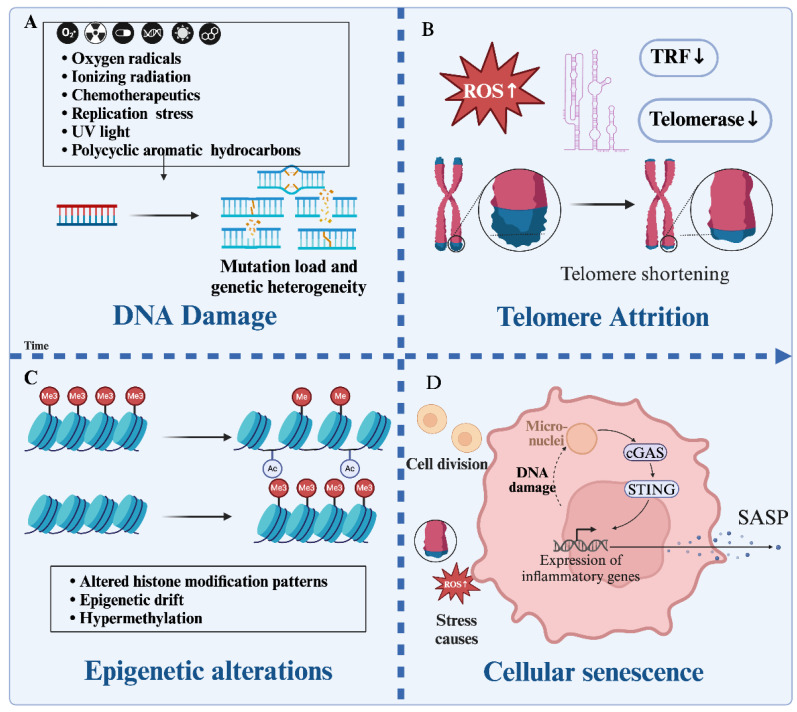
(**A**) DNA damage can arise from both endogenous sources (e.g., oxidative stress, replication errors) and exogenous sources (e.g., UV radiation, chemicals). The accumulation of DNA damage increases genomic instability and leads to a variety of chronic diseases. (**B**) Telomeres shorten as cells divide, when telomerase fails to repair them, or due to oxidative damage. (**C**) When becoming aged, epigenetic patterns deteriorate, leading to a loss of specificity and stochastic changes in gene expression that impair cellular function. (**D**) As cells continue to divide to their limits, or because of the induction of external stressful pressures, they move into senescence. Created with BioRender.com.

**Figure 4 cells-14-00619-f004:**
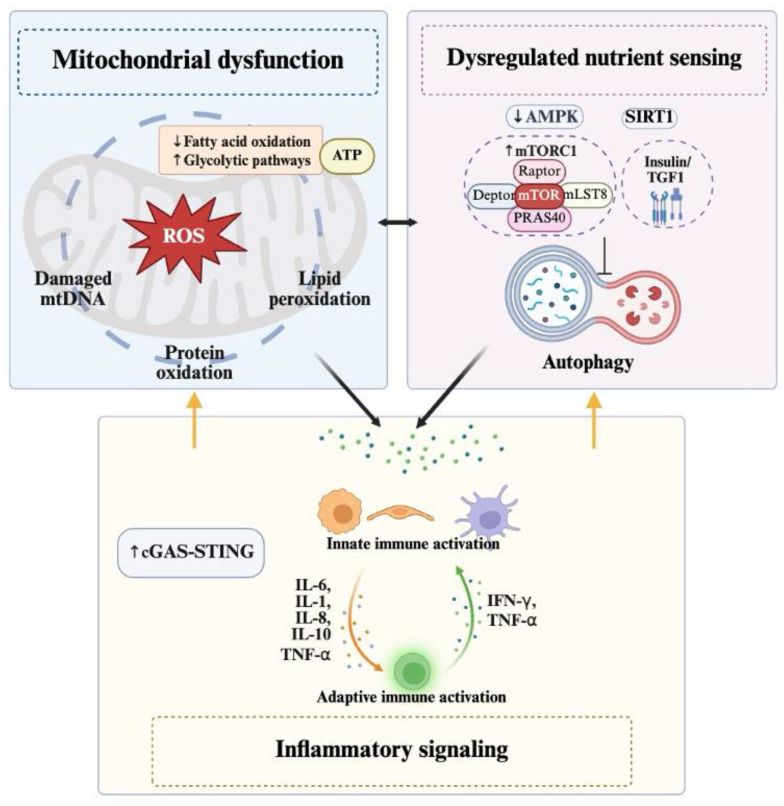
At the molecular level, mitochondrial dysfunction, dysregulated nutrient sensing, and inflammatory signaling are interconnected and together drive the aging process. With aging, mitochondrial function declines, and major ATP production patterns change, while exhibiting abnormalities such as DNA damage, increased inflammation, protein oxidation, and lipid peroxidation. The dysregulation of the nutrient sensing system, particularly the AMPK pathway and SIRT1 pathway, affects metabolism, while alterations in the mTOR pathway and insulin/TGF1 pathway inhibit autophagy. All of the above alterations cause the sustained reciprocal activation of the innate and adaptive immune systems in the environment, which, together with the activation of the cGAS–STING pathway, promote environmental inflammation. Created with BioRender.com.

**Figure 5 cells-14-00619-f005:**
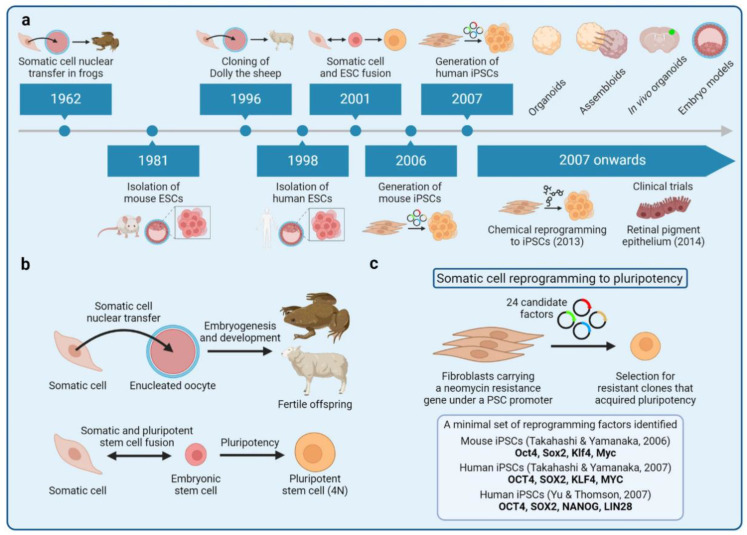
Development of the iPSC technology. Figure from [120], used under a Creative Commons Attribution 4.0 International License. Source: https://www.nature.com/articles/s41392-024-01809-0#Fig1 (accessed on 26 April 2024). The literature mentioned in the figure is from [14,121,122], no copyright issues. Development of the induced pluripotent stem cell (iPSC) technology. (**a**) A timeline of key breakthroughs related to the iPSC technology. (**b**) (Top) Somatic cell nuclear transfer (SCNT) experiments were pioneered by John Gurdon in the African clawed frog. Gurdon demonstrated that somatic cells retained all the genetic information necessary to give rise to a germline-competent organism. Successful SCNT in mammals was demonstrated by Keith Campbell, Ian Wilmut, and colleagues who cloned Dolly the sheep. (Bottom) Masako Tada and colleagues demonstrated that pluripotency can also be achieved by fusing a somatic cell with an embryonic stem cell, leading to the formation of a hybrid tetraploid cell. 4N, tetraploid. (**c**) The groundbreaking experiments of fibroblast reprogramming to pluripotency were pioneered by Kazutoshi Takahashi and Shinya Yamanaka. The researchers selected 24 factors as candidates for reprogramming and delivered these factors into mouse fibroblasts in various combinations by retroviral transduction. Eventually, Takahashi and Yamanaka identified a combination of 4 reprogramming factors—Oct4, Sox2, Klf4, and Myc—that was sufficient to reprogram mouse fibroblasts into embryonic stem cell-like pluripotent cells, known as iPSCs. Subsequently, Yamanaka and James Thomson independently reprogrammed human fibroblasts into iPSCs in 2007.

**Figure 6 cells-14-00619-f006:**
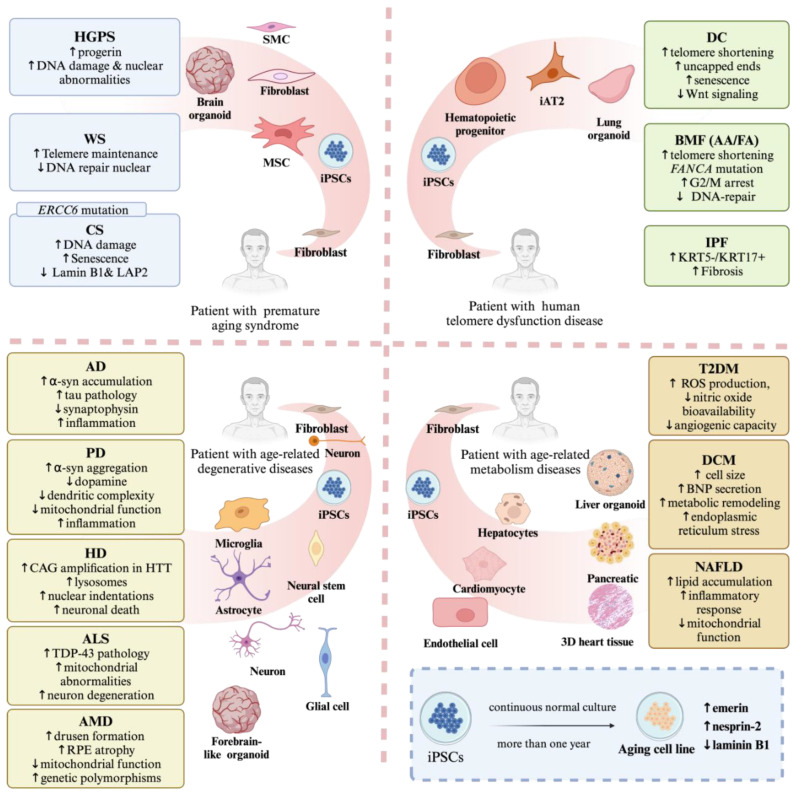
Induced pluripotent stem cell-based models for human aging diseases investigation. Somatic cells from patients with age-related diseases (including premature aging syndromes, telomere dysfunction disease, degenerative diseases, and metabolism diseases) can be induced into iPSCs, which in turn can differentiate into a desired cell type under various stimulated cultures, or even differentiate multiple cells to form organoids. These cells and organoids can restore the pathological manifestations of diseases, which can help researchers to investigate the process and causes of diseases. In addition to this, normal iPSCs also show some aging characteristics after long-term culture or aging induction. Created with BioRender.com.

**Figure 7 cells-14-00619-f007:**
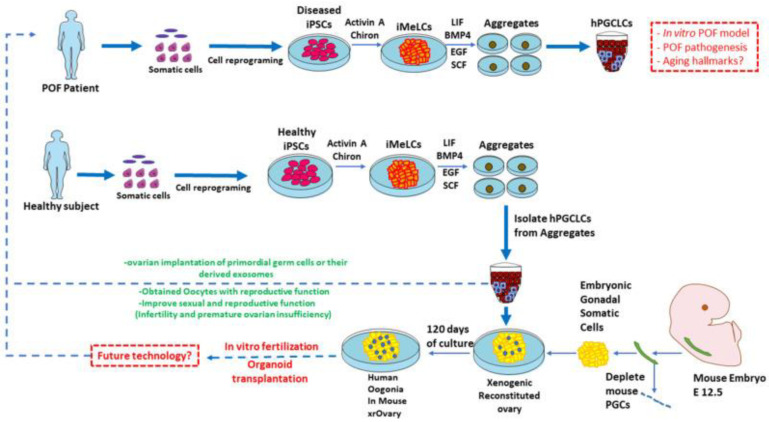
A schematic representation of the generation of primordial germ cells from POI patient cells for use as an in vitro POI model, and to generate human oogonia from healthy subject somatic cells as a future therapeutic option. Figure derived from [239], used under a Creative Commons Attribution 4.0 International License. Source: https://www.mdpi.com/2073-4409/11/23/3713, accessed on 26 April 2024.

**Figure 8 cells-14-00619-f008:**
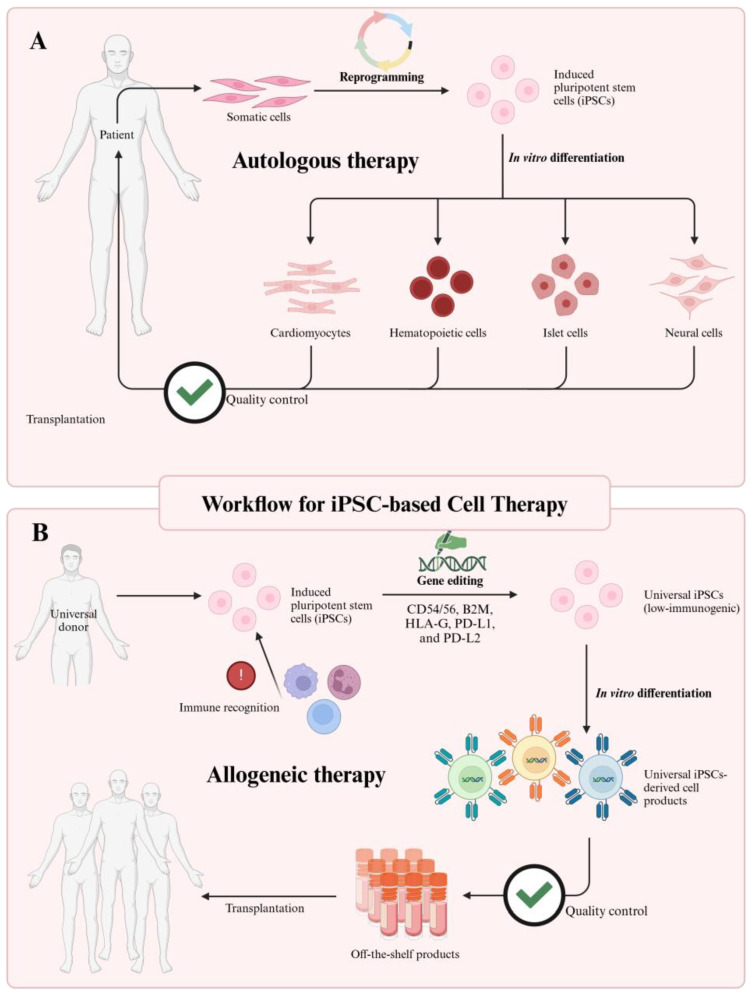
The transplantation therapy of iPSC can be categorized into (**A**) autologous transplantation and (**B**) allogeneic transplantation. Created with BioRender.com.

**Figure 9 cells-14-00619-f009:**
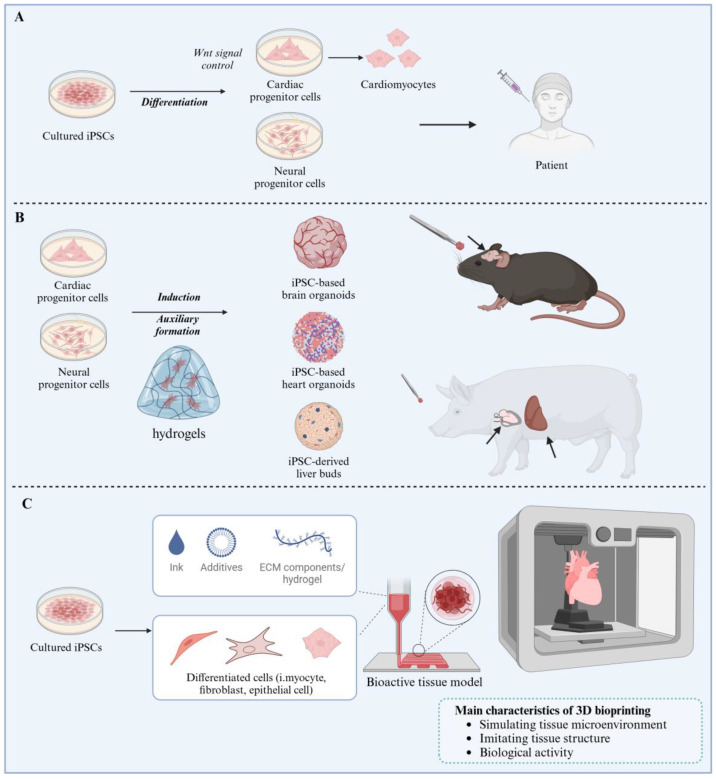
(**A**) iPSC-induced cell therapy has recently emerged as a promising approach to repair or replace damaged tissues. (**B**) In addition, iPSC-derived cells can be smoothly molded to form organoids with the help of other tissue-engineered structures, such as hydrogels, and transplanted into mouse and pig models, or even into patients. (**C**) Finally, iPSC-derived cells can be printed into these three-dimensional (3D) structures with the help of bioinks to form functional tissues. Created by BioRender.com.

**Figure 10 cells-14-00619-f010:**
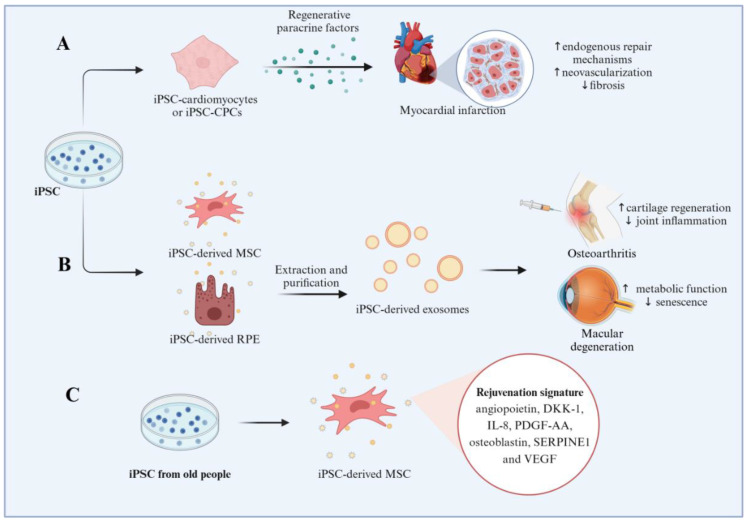
(**A**) The iPSC-derived cells can have beneficial effects on the infused organ as well as surrounding cells in the form of paracrine secretion. (**B**) Exosomes derived from iPSC-derived cells have been used to treat aging-related diseases such as osteoarthritis and macular degeneration. (**C**) In addition, iPSC-derived mesenchymal stem cells (MSCs) from older adults can acquire rejuvenating features relevant to regenerative medicine. Created with BioRender.com.

**Figure 11 cells-14-00619-f011:**
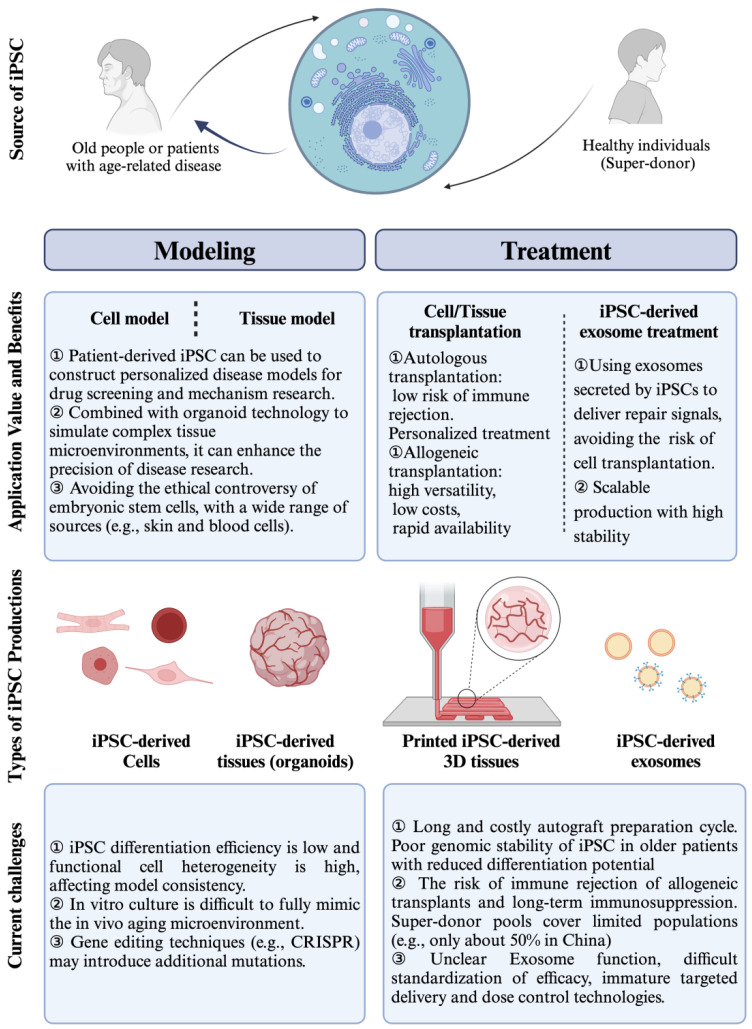
The source of iPSC can be categorized as either the aging population/patients with aging-related diseases or healthy individuals (Super-donors). iPSC can be used as either a model builder or in therapeutic approaches for aging or aging-related diseases. While iPSC has great application value in the study of aging diseases and is a very promising therapeutic approach, there are still various challenges, such as the sourcing and preparation of cells, as well as efficacy and safety. Created with BioRender.com.

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
