# Peer review of "Induced Pluripotent Stem Cells-Based Regenerative Therapies in Treating Human Aging-Related Functional Decline and Diseases"

_cells, 2025, doi:10.3390/cells14080619_

Round 1
Reviewer 1 Report
Comments and Suggestions for Authors
This review deals with putative applications of induced pluripotent stem cells in various age-related conditions. The review includes state-of-the-art technologies and current animal models as well as clinical studies on the subject. It is an interesting topic with a highly evolving impact in regenerative medicine as well. Some important points require attention as follows:
-The writing is too dense on occasions and full of acronyms (many of them undefined in the text), making it difficult to read. Several examples: 1) Paragraph 3.1 should be more elaborated to illustrate the putative role of OSKM factors in cancer, or the advantage of using iPSCs to generate genetically modified animal models; 2) The paragraph describing studies of the phenotype of motor neurons derived from iPSCs with TDP43 mutation is hard to read and confusing (lines 804-29); 3) The subheading 4.3.5. dealing with iPSCs as a model for macular degeneration needs clarification of the rationale concerning some of the described studies; 4) Fig. 6 should be properly described in detail; it is too general; 5) The last paragraph of subheading 5.5 is hard to follow.
- Some paragraphs seem to be misplaced, namely lines 447-51. Paragraph 3.3 might be merged into the previous one; and subheading 5.4 should be merged with 5.3.
- Some important descriptions are missing, namely insulin-resistant medium (line 938); the mechanism of rapamycin-activated caspase-9 in low-immunogenic hiPSC (lines 1029-30).
- Reference 155 deals with experiments using iPS-derived mesenchymal cells (p. 15). reference 79 is missing.
Author Response
|
Comments 1: The writing is too dense on occasions and full of acronyms (many of them undefined in the text), making it difficult to read. Several examples: 1) Paragraph 3.1 should be more elaborated to illustrate the putative role of OSKM factors in cancer, or the advantage of using iPSCs to generate genetically modified animal models; 2) The paragraph describing studies of the phenotype of motor neurons derived from iPSCs with TDP43 mutation is hard to read and confusing (lines 804-29); 3) The subheading 4.3.5. dealing with iPSCs as a model for macular degeneration needs clarification of the rationale concerning some of the described studies; 4) Fig. 6 should be properly described in detail; it is too general; 5) The last paragraph of subheading 5.5 is hard to follow.
|
|
Response 1: We thank you a lot for these comments, and these comments helped us a lot for improving the manuscript. We have included a dedicated table of abbreviations (or a glossary) to clearly define all acronyms used in the manuscript (Tite page, Line 31). And we carefully polished most of text to make words fluent, especially the parts you mentioned above. (1) For section 3.1 with the relationship between OSKM factors and cancer and adjusted the language to make it more fluent. Please refer to Page 12 Lines 396 for details. “Although OSKM factors are associated with tumor formation, their application under specific conditions offers great potential and benefits for scientific research and medicine. The process of reprogramming is initiated by introducing specific reprogramming factors into the target cells. These factors can activate or repress key genes responsible for maintaining pluripotency and play a key role in the reprogramming process [13].” (2) We revised and embellished the paragraphs describing the studies on the phenotype of iPSC-derived motor neurons with TDP43 mutations to make them more fluent. Please refer to Page 22 Lines 822 for details. “One of the most prominent features of ALS is the cytoplasmic mislocalization, aggregation, and phosphorylation of TDP-43 [214]. Using iPSCs derived from ALS patients, Ritsma, Laila et al. discovered that the aggregation of TDP-43 is often due to an impaired proteasomal degradation process. This impairment leads to the mis-splicing of the neuronal growth-associated factor stathmin 2 (STMN-2), resulting in axonal degeneration [214]. In further study, Lépine, S et al. utilized CRISPR/Cas9-engineered iPSC strains carrying the TDP-43 mutation and found different results. Interestingly, mutant motoneurons did not exhibit typical ALS pathological features associated with TDP-43, such as obvious aggregation, increased phosphorylation, and abnormal nuclear-cytoplasmic distribution. Instead, they showed a gradual weakening of spontaneous neural activity, characterized by reduced electrophysiological function, and abnormalities in synaptic structure and function [211].” (3) We embellished and modified the language of this passage to make it more fluent. Please refer to Page 23 Lines 880 for details. “Notably, these models revealed that AMD high-risk alleles (CFH and ARMS2) disrupt RPE pigmentation by activating NF-κB signaling while suppressing autophagy pathways [221]. Through high-throughput drug screening, Sharma et al. identified L-745,870 (a dopamine antagonist) and aminocaproic acid (a protease inhibitor) as compounds that normalize RPE pigmentation and epithelial morphology even in CFH(H/H) genotype cells [221].” (4) Thanks for pointing this out. We have updated the figure notes for Figure 6 to make them more detailed and understandable. Please refer to Page 16 Lines 545 or details. “Figure 6. Induced pluripotent stem cell-based models for human aging diseases investigation. Somatic cells from patients with age-related diseases (including premature aging syndromes, telomere dysfunction disease, degenerative diseases, and metabolism diseases) can be induced into iPSCs, which in turn can differentiate into a desired cell type under various stimulated cultures, or even differentiate multiple cells to form organoids. These cells and organoids can restore the pathological manifestations of diseases, which can help researchers to investigate the process and causes of diseases. In addition to this, normal iPSCs also show some aging characteristics after long-term culture or aging induction.” (5) We appreciate your suggestion. We have revised section 5.5 to iPSC-induced immunotherapy for anti-aging to better emphasize the anti-aging theme. In addition, potential methods to improve the quality of iPSC production we have added in section 3. Please refer to Page 33 Lines 1207 and Page 14 Line 489 for details.
|
|
Comments 2: Some paragraphs seem to be misplaced, namely lines 447-51. Paragraph 3.3 might be merged into the previous one; and subheading 5.4 should be merged with 5.3. |
|
Response 2: Thanks for pointing this out. We made modifications for lines 447-451. “Since their discovery two decades ago, iPSC generation has experienced exponential growth in research applications. However, the extended timeline required for functional human neuronal differentiation, coupled with the constraints of conventional 2D culture systems, continues to present substantial hurdles for disease modeling and therapeutic development. To address these challenges, researchers have turned to innovative approaches like conductive graphene scaffolds, which enhance neuronal differentiation by simultaneously providing mechanical support and electrical stimulation [134].” We also changed the order of statements to make the paragraphs more coherent. Please refer to Page 13 Lines 453 for details. (1) Paragraph 3.3 was merged with the previous paragraph. Please refer to Page 13 Lines 424 for details. (2) We changed the subheading 5.4 for better visualization. This is because there is a clear distinction between this section and the previous one, i.e., iPSC produces beneficial anti-aging effects by paracrine means. Please refer to Page 31 Lines 1170 for details.
|
|
Comments 3: Some important descriptions are missing, namely insulin-resistant medium (line 938); the mechanism of rapamycin-activated caspase-9 in low-immunogenic hiPSC (lines 1029-30). |
|
Response 3: Thanks for pointing this out. These important notes are very important for understanding the text, and we added the necessary sentences. Please refer to Page 24 Lines 936 and Page 27 Lines 1022 for details. “In a similar study by Carter et al, human iPSC-CMs successfully induced insulin resistance and mimicked the metabolic features and diastolic dysfunction of T2DM after being cultured for 6 days in medium with or without insulin, respectively. Importantly, HyPSC-derived HPCs had the ability to evade both innate and adaptive immune responses, and even rapamycin-activated caspase 9 serves as its safety switch [237].”
|
|
Comments 4: Reference 155 deals with experiments using iPS-derived mesenchymal cells (p. 15). reference 79 is missing. |
|
Response 4: Thank you for pointing out. We made up for missed text and deleted missing irrelevant literature. Please refer to Page 17 Lines 579 for details. |
Reviewer 2 Report
Comments and Suggestions for Authors
The authors offer a thorough review of the cellular and molecular hallmarks of age-related decline and associated diseases, and detailed overview of how existing stem cell therapies can help in the battle against age-related pathologies.
Main comments
The review is comprehensive and thorough, reads fluently in most parts. The English is good overall, some minor errors exist, so needs checking by the publisher. Some paragraphs and sections have less confluence in the sentences that follow each other and I have marked these in the specific comments.
One important topic, which is mentioned in a few places but not explained or detailed, is age rejuvenation of iPSCs and their derived differentiated progeny – i.e. iPSCs are also age-reprogrammed, their biological age is brought back to ‘zero’ (technically, slightly less than zero) and the cells differentiated from iPSC retain younger biological age than the actual age of the original primary cell donor. This is an obvious drawback specifically when using iPSCs-derived cells to study age-related disease (especially when time-bound ‘aging’ phenotype is sought rather than a mutation phenotype), so it needs special attention. On the other hand, this is an entirely novel avenue for regenerative medicine, when the transplanted cells are purposefully made epigenetically ‘younger’. Several places in the MS would benefit in detailing this topic a bit more for the reader – some examples pointed below.
Some inconsistencies between sentences and sentences without much sense here and there, some very fine details ‘out of the blue’ about studies here and there as if just copied from somewhere without explanation or context - leaves the impression of heavy use of generative AI in the preparation of this review, therefore the MS should be edited and thoroughly checked by the authors and checked against MDPI’s policies.
Specific and minor comments:
Line 107 – rejuvenation is a very hot and emerging field, it is appropriate to add a citation about successful rejuvenation of stem cells
Line – 321-322 – “liver-specific IGF-1 deficiency led to impaired hepatic function and autophagy in aged mice” > is ‘impaired’ a typo from ‘improved’, because the study shows improvement and attenuation of aging specific decline with role of autophagy
Line 406 – “OSKM can be used to induce and maintain pluripotency in mice” – in mouse cells, or transgenic mice? unclear sentence
Lines 406 – 412 – every sentence is unclear and not logically making sense – those should be checked and corrected, also the connection between the sentences is not very smooth. The last two sentences refer to tumors and need editing to logically flow from the previous sentences in the paragraph, and also mention an OSKM ‘gene’ (“the OSKM gene correlates with patient prognosis”), which is not accurate for patients, unless talking about transgenic animals with a polycistronic OSKM gene etc.
Lines 458-461 – this sentence (about Yu J et al) has no connection to the paragraph, which discusses viral integration, while this sentence is about not using feeders (mouse fibroblast layer) in an iPSC culture. It should be removed and placed in a more suitable context.
Lines 464-482 – this entire paragraph goes into very high detail about mechanical cues, without explaining and giving background – as if all readers are experts in biomechanics, which they most likely won’t be. It should be explained as a normal review would do, why is it done and what is the benefit, before going into such detail that is actually irrelevant.
Line 566-576 - “retain the ability to re-differentiate into fully rejuvenated cells” > this sentence mentions rejuvenation, which is an important phenomenon that characterises iPSCs and has not been detailed so far, but is mentioned as common knowledge. This needs its own paragraph earlier on with respective citations as mentioned in main comments.
Lines 663 – 665 – “Indeed, early cellular models using human iPSCs for AD and tau proteinopathies only recapitulated the early stages of the disease and had difficulty to show protein aggregation.” > human iPSCs-derived neurons cannot recapitulate a fully mature epigenome and hence mature fully functionally, maybe this is important to mention here, see Martin, Poppe et al (https://doi.org/10.3390/genes14050957). Again linked to age-reprograming of iPSC-derived cells.
Line 776 – correct gene name: huntingtin (HTT)
Line 809 – “A further study by Lépine, S et al. using CRISPR/Cas9 engineered iPSC strains carrying the TDP-43 mutation found different [203].” > sentence unfinished?
Lines 1125-6 – again rejuvenation mentioned, please see above comment and devote a paragraph with explanation of the positives or negatives of age-reprogrammed (Rejuvenated) iPSC-derived cells.
Line 1173 - 5.4. Mitigating aging-related functional decline. > this paragraph should explain why these MSCs are rejuvenated and that it is a common phenomenon for all iPSC-derived cells from healthy aged cells. All described iPSC-derived cells would have rejuvenated properties, however when disease-associated mutations are present, they naturally dominate the phenotype, but these cells would also lack some of the age-related properties.
Lines 1232-4 – again very detailed about technology and terminology that would be understood by very niche specialists – “lipid insertion displays DNA on the cell surface faster and more efficiently than click binding” - what is ‘lipid insertion’ or ‘click-binding’, it is not mentioned elsewhere, why such detail? Can the authors explain what the ultimate take-home message from that experiment/cited reference is? This looks like copying sentences from an abstract and just putting them there without making the effort to explain to the reader why this is relevant and what it ultimately means.
Comments on the Quality of English LanguageEnglish is fine, some minor check and editing is required.
Author Response
|
Comments 1: One important topic, which is mentioned in a few places but not explained or detailed, is age rejuvenation of iPSCs and their derived differentiated progeny – i.e. iPSCs are also age-reprogrammed, their biological age is brought back to ‘zero’ (technically, slightly less than zero) and the cells differentiated from iPSC retain younger biological age than the actual age of the original primary cell donor. This is an obvious drawback specifically when using iPSCs-derived cells to study age-related disease (especially when time-bound ‘aging’ phenotype is sought rather than a mutation phenotype), so it needs special attention. On the other hand, this is an entirely novel avenue for regenerative medicine, when the transplanted cells are purposefully made epigenetically ‘younger’. Several places in the MS would benefit in detailing this topic a bit more for the reader – some examples pointed below. Response 1: Thanks a lot for these valuable comments. We made additional demonstration in section 4 (Page 15 Line 526): “Here we want to emphasize that the process of generating iPSCs involves a phenomenon called age reprogramming, which resets the biological age of cells to a younger state. This rejuvenation effect is evident in various cellular characteristics, including gene expression patterns, and epigenetic markers [149,150]. This age reversal is a key characteristic of iPSC technology and has significant implications for both basic research and potential therapeutic applications. While it allows for the study of early developmental processes and the generation of unlimited cell sources, it can also complicate the modeling of late-onset diseases, especially when time-dependent aging phenotype is sought, as the rejuvenated cells may not accurately reflect the aged state in which these diseases typically manifest.”, and section 5 (Page 25 Line 989): “Aging-related diseases often involve the dysfunction or loss of specific cell types, leading to organ and tissue degeneration. Due to their "young" characteristics, iPSCs offer a promising solution by enabling the reprogramming of adult cells into a pluripotent state,”. |
|
|
|
Comments 2: Some inconsistencies between sentences and sentences without much sense here and there, some very fine details ‘out of the blue’ about studies here and there as if just copied from somewhere without explanation or context - leaves the impression of heavy use of generative AI in the preparation of this review, therefore the MS should be edited and thoroughly checked by the authors and checked against MDPI’s policies. |
|
Response 2: Thanks for pointing this out. We carefully polished most texts to make words fluent and checked the AI in the MS.
|
|
Comments 3: Line 107 – rejuvenation is a very hot and emerging field, it is appropriate to add a citation about successful rejuvenation of stem cells. |
|
Response 3: Thanks a lot for this comment. We made additional demonstration about “rejuvenation” in section 4 (Page 15 Line 526), which was also responded in Response 1.
|
|
Comments 4: Line – 321-322 – “liver-specific IGF-1 deficiency led to impaired hepatic function and autophagy in aged mice” > is ‘impaired’ a typo from ‘improved’, because the study shows improvement and attenuation of aging specific decline with role of autophagy |
|
Response 4: Thank you for pointing this out. We revised that sentence (Page 10 Lines 322): “Studies focusing on cardiac aging also found that overexpression of IGF-1 improved cardiomyocyte contractile function in old mice”.
|
|
Comments 5: Line 406 – “OSKM can be used to induce and maintain pluripotency in mice” – in mouse cells, or transgenic mice? unclear sentence |
|
Response 5: We agree with this comment. Thus, we complemented the methods and conclusions of this study to highlight the importance and potential application of OSKM-induced iPSC generation in Page 12 Lines 410: “The mouse cell line established by Mao et al. consistently induces OSKM expression to derive and maintain pluripotent cells without the use of specific growth factors or signaling inhibitors that are typically required in conventional culture systems. Importantly, further experiments injecting iPSC generated by sustained OSKM expression into mouse blastocysts revealed that OSKM-iPSC contributed to various organs and tissues after the formation of chimeric embryos. This study showed that OSKM-induced iPSCs have the potential for application in other species to generate genetically modified animals through lineage transmission.”.
|
|
Comments 6: Lines 406 – 412 – every sentence is unclear and not logically making sense – those should be checked and corrected, also the connection between the sentences is not very smooth. The last two sentences refer to tumors and need editing to logically flow from the previous sentences in the paragraph, and also mention an OSKM ‘gene’ (“the OSKM gene correlates with patient prognosis”), which is not accurate for patients, unless talking about transgenic animals with a polycistronic OSKM gene etc. |
|
Response 6: Thank you for pointing this out. To make the logic of the paragraph flow better, we placed the reference to cancer at the beginning (Page 12 Lines 410): “The mouse cell line established by Mao et al. consistently induces OSKM expression to derive and maintain pluripotent cells without the use of specific growth factors or signaling inhibitors that are typically required in conventional culture systems. Importantly, further experiments injecting iPSC generated by sustained OSKM expression into mouse blastocysts revealed that OSKM-iPSC contributed to various organs and tissues after the formation of chimeric embryos. This study showed that OSKM-induced iPSCs have the potential for application in other species to generate genetically modified animals through lineage transmission.”.
|
|
Comments 7: Lines 458-461 – this sentence (about Yu J et al) has no connection to the paragraph, which discusses viral integration, while this sentence is about not using feeders (mouse fibroblast layer) in an iPSC culture. It should be removed and placed in a more suitable context. |
|
Response 7: Thanks to your suggestion, we revised the paragraph and added appropriate examples at Page 13 Lines 466: “As an RNA virus, Sendai virus replicates in the cytoplasm and does not integrate into the host cell's genome, thus avoiding insertional mutagenesis [137]. Episomal vectors are maintained as extrachromosomal elements and are gradually lost during cell division, and thus offer a safer alternative to integrating viral vectors, reducing the risk of insertional mutagenesis and oncogene activation [136].”.
|
|
Comments 8: Lines 464-482 – this entire paragraph goes into very high detail about mechanical cues, without explaining and giving background – as if all readers are experts in biomechanics, which they most likely won’t be. It should be explained as a normal review would do, why is it done and what is the benefit, before going into such detail that is actually irrelevant. |
|
Response 8: Thank you for this valuable comment. We made additional modification in Page14 Lines 473 accordingly: “Importantly, due to the potential carcinogenic risk of OSKM, there are ongoing efforts to replace or supplement these factors with extracellular signals and microenvironmental cues, especially those synthetic biomaterials. The stiffness of the substrate was found to guide stem cell differentiation into specific lineages, such as neurons or muscle cells, while softer substrates have been shown to improve human cell reprogramming outcomes compared to stiffer ones [138-140]. For example, Chowdhury et al. found that a novel reprogramming regulator - protein phosphatase and actin regulatory factor 3 - is upregulated at very early time points under gel conditions, which accounts for the enhanced reprogramming results observed [140].”.
|
|
Comments 9: Line 566-576 - “retain the ability to re-differentiate into fully rejuvenated cells” > this sentence mentions rejuvenation, which is an important phenomenon that characterises iPSCs and has not been detailed so far, but is mentioned as common knowledge. This needs its own paragraph earlier on with respective citations as mentioned in main comments. |
|
Response 9: Thank you for this comment. We made additional demonstrations We made additional demonstration in section 4 (Page 15 Line 526): “Here we want to emphasize that the process of generating iPSCs involves a phenomenon called age reprogramming, which resets the biological age of cells to a younger state. This rejuvenation effect is evident in various cellular characteristics, including gene expression patterns, and epigenetic markers [149,150]. This age reversal is a key characteristic of iPSC technology and has significant implications for both basic research and potential therapeutic applications. While it allows for the study of early developmental processes and the generation of unlimited cell sources, it can also complicate the modeling of late-onset diseases, especially when time-dependent aging phenotype is sought, as the rejuvenated cells may not accurately reflect the aged state in which these diseases typically manifest.”, and section 5 (Page 25 Line 989): “Aging-related diseases often involve the dysfunction or loss of specific cell types, leading to organ and tissue degeneration. Due to their "young" characteristics, iPSCs offer a promising solution by enabling the reprogramming of adult cells into a pluripotent state,”.
|
|
Comments 10: Lines 663 – 665 – “Indeed, early cellular models using human iPSCs for AD and tau proteinopathies only recapitulated the early stages of the disease and had difficulty to show protein aggregation.” > human iPSCs-derived neurons cannot recapitulate a fully mature epigenome and hence mature fully functionally, maybe this is important to mention here, see Martin, Poppe et al (https://doi.org/10.3390/genes14050957). Again linked to age-reprograming of iPSC-derived cells. |
|
Response 10: Thanks a lot for this comment, we made additional demonstration in Page 19 Lines 682: “Indeed, early cellular models using human iPSCs for AD and tau proteinopathies only recapitulated the early stages of the disease and had difficulty to show protein aggregation, which may be related to the fact that the reprogramming process of iPSCs causes it to go to a “young” state as mentioned before. And unlike human mES-derived neurons, Sally Martin et al. found that mouse ESC-derived neurons were able to recapitulate the unique DNA methylation profiles of adult neurons within a controlled timeframe of in vitro experiments, which allows them to serve as a model system for studying epigenome maturation during development [190].”.
|
|
Comments 11: Line 776 – correct gene name: huntingtin (HTT) |
|
Response 11: Thanks for pointing out. We have corrected this gene name in Page 21 Lines 795: “HD is a neurodegenerative late-onset genetic disorder caused by a CAG amplification in the coding region of the Huntingtin gene, resulting in toxic polyglutamine stretches in the huntingtin protein.”.
|
|
Comments 12: Line 809 – “A further study by Lépine, S et al. using CRISPR/Cas9 engineered iPSC strains carrying the TDP-43 mutation found different [203].” > sentence unfinished? |
|
Response 12: Thanks for pointing out. We added additional demonstration in Page 22 Lines 822: “In a further study, Lépine, S et al. utilized CRISPR/Cas9-engineered iPSC strains carrying the TDP-43 mutation and found different results. Interestingly, mutant motoneurons did not exhibit typical ALS pathological features associated with TDP-43, such as obvious aggregation, increased phosphorylation, and abnormal nuclear-cytoplasmic distribution. Instead, they showed a gradual weakening of spontaneous neural activity, characterized by reduced electrophysiological function, and abnormalities in synaptic structure and function [211]”.
|
|
Comments 13: Lines 1125-6 – again rejuvenation mentioned, please see above comment and devote a paragraph with explanation of the positives or negatives of age-reprogrammed (Rejuvenated) iPSC-derived cells. |
|
Response 13: Thank you for this comment. We made additional demonstrations in section 4 (Page 15 Line 526) and section 5 (Page 25 Line 989).
|
|
Comments 14: Line 1173 - 5.4. Mitigating aging-related functional decline. > this paragraph should explain why these MSCs are rejuvenated and that it is a common phenomenon for all iPSC-derived cells from healthy aged cells. All described iPSC-derived cells would have rejuvenated properties, however when disease-associated mutations are present, they naturally dominate the phenotype, but these cells would also lack some of the age-related properties. |
|
Response 14: Thank you for this comment. We made additional demonstrations in section 4 (Page 15 Line 526) and section 5 (Page 25 Line 989).
|
|
Comments 15: Lines 1232-4 – again very detailed about technology and terminology that would be understood by very niche specialists – “lipid insertion displays DNA on the cell surface faster and more efficiently than click binding” - what is ‘lipid insertion’ or ‘click-binding’, it is not mentioned elsewhere, why such detail? Can the authors explain what the ultimate take-home message from that experiment/cited reference is? This looks like copying sentences from an abstract and just putting them there without making the effort to explain to the reader why this is relevant and what it ultimately means. |
|
Response 15: Thanks for pointing out this. We modified the content of this article and moved it to section 3 (Page 14 Line 489) to make it more logical: “Synthetic DNA combined with cell surface engineering can also address limitations related to iPSC genomic stability and reprogramming efficiency. For instance, introducing synthetic mRNA encoding CYCLIN D1, a protein involved in homologous recombination, has been shown to enhance DNA repair during reprogramming, leading to genetically stable iPSCs [143]. Importantly, Wang X et al. found that the physical method of embedding DNA into the cell membrane by combining it with lipids is faster and more efficient. The chemical reaction that covalently binds DNA to the cell surface can make the survival time (half-life) of DNA on the cell surface 3-4 times longer than that of DNA connected by physical methods, which has guiding significance for the surface engineering of iPSCs [144].”.
|
|
|

Reviewer 3 Report
Comments and Suggestions for Authors
In this review, the authors have broadly summarized the literature of aging-related studies with use of human Induced pluripotent stem cells (iPSCs) and praised the revolutionary role of iPSCs in aiding the aging-related research.
Indeed, research using the iPSC technology has accelerated our understanding of the underlying mechanism(s) of aging-related functional decline and diseases as well as paved the way for regenerative medicine for their treatment. However, current challenges of the translation of iPSC-based regenerative medicine to clinical treatment are still critical issues that need to be addressed, such as those mentioned by the authors including the tumorigenicity, immunogenicity, genetic instability and heterogeneity of the iPSCs after differentiation.
With respect to the above, the authors did not provide deeper insights to discuss more about the potential strategies and emerging knowledge dealing with these challenges of iPSC-based regenerative medicine.
The following points are also suggested for the authors to consider during revision of the manuscript for Cells:
(1) The title of this review article could be modified to Induced Pluripotent Stem Cells-based Regenerative medicine for treatment of Human Aging-Related Functional Decline and Diseases;
(2) As pointed out by the authors, autologous transplantation by reprograming patients’ own somatic cells into iPSCs and then differentiating them into the desired cell type(s) for therapy is one of the primary ways to mitigate immune rejection.
However, there is a missing link for the readers to understand how iPSCs-derived from aged patients could be further differentiated into healthy cells for the treatment of their aging associated diseases, since the accumulation of many genetic alterations and/or cellular malfunctions during aging has been emphasized by the authors in the early sections. The authors may want to add a few sentences on this point.
(3) Information of two other parts could be considered to be included in the review. First, the rejuvenation powers of young blood/ cells/ factors/ EVs (extracellular vesicles) described by previous studies could be briefly summarized. Second, emerging technologies/strategies of genetic manipulation and cellular engineering approaches for improvement of the quality/ function/ efficacy of cell therapy could also be reviewed briefly.
(4) Text writing, figures, references, etc.
(A) Full names of many abbreviations are missing in the text;
(B) Typos and duplicated/ incomplete sentences need to be corrected;
(C) Figure 4 is neither well-organized nor informative;
(D) The reference style used in this manuscript is chaotic and not reader friendly. In particular, many of the references are missing the journal names and/or publication time.
Author Response
|
Comments 1: The title of this review article could be modified to Induced Pluripotent Stem Cells-based Regenerative medicine for treatment of Human Aging-Related Functional Decline and Diseases |
|
Response 1: We thank this reviewer for this comment. We changed the title as you suggested “Induced Pluripotent Stem Cells-based Regenerative Therapies in Treating Human Aging-Related Functional Decline and Diseases”.
|
|
Comments 2: As pointed out by the authors, autologous transplantation by reprograming patients’ own somatic cells into iPSCs and then differentiating them into the desired cell type(s) for therapy is one of the primary ways to mitigate immune rejection. However, there is a missing link for the readers to understand how iPSCs-derived from aged patients could be further differentiated into healthy cells for the treatment of their aging associated diseases, since the accumulation of many genetic alterations and/or cellular malfunctions during aging has been emphasized by the authors in the early sections. The authors may want to add a few sentences on this point. |
|
Response 2: Thanks for pointing this out. We made additional demonstrations in section 4, Page 15 Line 526-Line 535: “Here we want to emphasize that the process of generating iPSCs involves a phenomenon called age reprogramming, which resets the biological age of cells to a younger state. This rejuvenation effect is evident in various cellular characteristics, including gene expression patterns, and epigenetic markers [149,150]. This age reversal is a key characteristic of iPSC technology and has significant implications for both basic research and potential therapeutic applications. While it allows for the study of early developmental processes and the generation of unlimited cell sources, it can also complicate the modeling of late-onset diseases, especially when time-dependent aging phenotype is sought, as the rejuvenated cells may not accurately reflect the aged state in which these diseases typically manifest.” and section 5, Page 25 Line 989: “Aging-related diseases often involve the dysfunction or loss of specific cell types, leading to organ and tissue degeneration. Due to their "young" characteristics, iPSCs offer a promising solution by enabling the reprogramming of adult cells into a pluripotent state,”.
|
|
Comments 3: Information of two other parts could be considered to be included in the review. First, the rejuvenation powers of young blood/ cells/ factors/ EVs (extracellular vesicles) described by previous studies could be briefly summarized. Second, emerging technologies/strategies of genetic manipulation and cellular engineering approaches for improvement of the quality/ function/ efficacy of cell therapy could also be reviewed briefly. |
|
Response 3: Thanks a lot for this valuable suggestion. We revised the title of 5.4 (Page 31 Lines 1170) into “Therapeutic paracrine secretion”. And we also revised the title of 5.5 (Page 33 Lines 1207) into “iPSC-induced immunotherapy for anti-aging”. In addition, we made additional demonstrations about the potential methods to improve the quality of iPSC production in Section 3 (Page 14 Line 489): “Synthetic DNA combined with cell surface engineering can also address limitations related to iPSC genomic stability and reprogramming efficiency. For instance, introducing synthetic mRNA encoding CYCLIN D1, a protein involved in homologous recombination, has been shown to enhance DNA repair during reprogramming, leading to genetically stable iPSCs [143]. Importantly, Wang X et al. found that the physical method of embedding DNA into the cell membrane by combining it with lipids is faster and more efficient. The chemical reaction that covalently binds DNA to the cell surface can make the survival time (half-life) of DNA on the cell surface 3-4 times longer than that of DNA connected by physical methods, which has guiding significance for the surface engineering of iPSCs [144].”.
|
|
Comments 4: (A) Full names of many abbreviations are missing in the text; (B) Typos and duplicated/ incomplete sentences need to be corrected; (C) Figure 4 is neither well-organized nor informative; (D) The reference style used in this manuscript is chaotic and not reader friendly. In particular, many of the references are missing the journal names and/or publication time. |
|
Response 4: Thanks a lot for these valuable comments. We made additional modifications according to these valuable comments: (A) We prepared a list of acronyms as an attachment (Page 1 Line 31). (B) We carefully checked the manuscript and revised the typos and duplicated/ incomplete sentences. (C) We revised Fig. 4 to make it more clear and organized.
(D) We reworked the citation insertion and updated the proper citation format (Page 34 Line 1276).
|
|
|
|
|

Round 2
Reviewer 1 Report
Comments and Suggestions for Authors
This revised version of the present review has significantly improved after the authors appropriately addressed my previous concerns.